



# Assessing impacts of selective logging on water, energy, and carbon budgets and ecosystem dynamics in Amazon forests using the Functionally Assembled Terrestrial Ecosystem Simulator

Maoyi Huang[1*], Yi Xu[1,2], Marcos Longo[3,4], Michael Keller[3,4,5], Ryan Knox[6], Charles Koven[6], Rosie Fisher[7]

[1]Atmospheric Sciences and Global Change Division, Pacific Northwest National Laboratory, Richland, WA, USA
[2]School of Geography, Nanjing Normal University, Nanjing, China
[3]Embrapa Agricultural Informatics, Campinas, SP, Brazil
[4]Jet Propulsion Laboratory, California Institute of Technology, Pasadena, CA, USA
[5]International Institute of Tropical Forestry, USDA Forest Service, Rio Piedras, Puerto Rico, USA
[6]Earth & Environmental Sciences Division, Lawrence Berkeley National Laboratory, Berkeley, CA, USA
[7]Climate and Global Dynamics Laboratory, National Center for Atmospheric Research, Boulder, CO, USA

*Correspondence to*: Maoyi Huang (Maoyi.Huang@pnnl.gov)





## Abstract

Tropical forest degradation from logging, fire, and fragmentation not only alters carbon stocks and carbon fluxes, but also impacts physical land-surface properties such as albedo and roughness length. Such impacts are poorly quantified to date due to difficulties in accessing and maintaining observational infrastructures, and the lack of proper modeling tools for capturing the interactions among biophysical properties, ecosystem demography, canopy structure, and biogeochemical cycling in tropical forests. As a first step to address these limitations, we implemented a selective logging module into the Functionally Assembled Terrestrial Ecosystem Simulator (FATES) by mimicking the ecological, biophysical, and biogeochemical processes following a logging event. The model can specify the timing and aerial extent of logging events, splitting the logged forest patch into disturbed and intact patches, determine the survivorship of cohorts in the disturbed patch, and modifying the biomass and necromass (total mass of coarse woody debris and litter) pools following logging. We parameterized the logging module to reproduce a selective logging experiment at the Tapajós National Forest in Brazil and benchmarked model outputs against available field measurements. Our results suggest that the model permits the coexistence of early and late successional functional types and realistically characterizes the seasonality of water and carbon fluxes and stocks, the forest structure and composition, and the ecosystem succession following disturbance. However, the current version of FATES overestimates water stress in the dry season therefore fails to capture seasonal variation in latent and sensible heat fluxes. Moreover, we observed a bias towards low stem density and leaf area when compared to observations, suggesting that improvements are needed in both carbon allocation and establishment of trees. The effects of logging were assessed by different logging scenarios to represent reduced impact and conventional logging practices, both with high and low logging intensities. The model simulations suggest that in comparison to old-growth forests the logged forests rapidly recover water and energy fluxes in one to three years. In contrast, the recovery times for carbon stocks, forest structure and composition are more than 30 years depending on logging practices and intensity. This study lays the foundation to simulate land use change and forest degradation in FATES, which will be an effective tool to directly represent forest management practices and regeneration in the context of Earth System Models.



## 1   Introduction

Land cover and land use in tropical forest regions are highly dynamic, and nearly all tropical forests are subject to significant human influence (*Martínez-Ramos et al.*, 2016;*Dirzo et al.*, 2014).  While old-growth tropical forests have been reported to be carbon sinks that remove carbon dioxide from the atmosphere through photosynthesis, these forests could easily become carbon sources once disturbed (*Luyssaert et al.*, 2008). Using data from forest inventory and long-term ecosystem carbon studies from 1990 to 2007, *Pan et al.* (2011) suggested a net tropical forest land-use source of $1.3 \pm 0.7$ Pg C $yr^{-1}$, consisting of a gross tropical deforestation loss of $2.9 \pm 0.5$ Pg C $yr^{-1}$ that is partially offset by a carbon uptake by tropical secondary forest regrowth of $1.6 \pm 0.5$ Pg C $yr^{-1}$. These estimates, however, do not account for tropical forest that has been degraded through the combined effects of selective logging (cutting and removal of merchantable timber), fuelwood harvest, understory fires, and fragmentation (*Nepstad et al.*, 1999;*Bradshaw et al.*, 2009). To date, the effects of forest degradation remain poorly quantified. Recent studies suggested that degradation may contribute to carbon loss 40% as large as clear cut deforestation (*Berenguer et al.*, 2014), and the emission from selective logging alone could be equivalent to ~10% to 50% of that from deforestation in the tropical countries (*Pearson et al.*, 2014;*Huang and Asner*, 2010;*Asner et al.*, 2009).  Selective logging of tropical forests is as an important contributor to many local and national economies, and correspond to approximately one eighth of global timber (*Blaser et al.*, 2011).  The integrated impact of timber production and other forest uses has been posited as the cause of up to ~30% of the difference between potential and actual biomass stocks globally, comparable in magnitude to the effects of deforestation (Erb et al. (2017).

Over half of all tropical forests have been cleared or logged, and almost half of standing old-growth tropical forests are designated by national forest services for timber production (Sist et al., 2015). Disturbances that result from logging are known to cause forest degradation at the same magnitude as deforestation each year in terms of both geographic extent and intensity, with widespread collateral damage to remaining trees, vegetation and soils, leading to disturbance to water, energy, and carbon cycling, as well as ecosystem integrity (*Keller et al.*, 2004b;*Asner et al.*, 2004;*Huang and Asner*, 2010).

In most Earth system models (ESMs) that couple terrestrial to atmospheric processes to investigate global change selective logging is typically represented as simple fractions of affected area or an amount of carbon to be removed on a coarse grid (e.g., 0.5 degree).  One exception is





the representation wood harvest in the LM3V land model that explicitly accounts for post-disturbance land age distribution, as part of the Geophysical Fluid Dynamics Laboratory (GFDL) Earth system model (*Shevliakova* et al., 2009). Grid cell fractional areas are typically based on timber production rates estimated from sawmill, sales, and export statistics (*Hurtt et al.*, 2011;*Lawrence et al.*, 2012). This approach, while practical, does not effectively differentiate selective logging that retains forest cover from deforestation. Selective logging includes cutting large trees and additional degradation through widespread damage to remaining trees, sub-canopy vegetation, and soils (*Asner et al.*, 2004;*Asner et al.*, 2005). Selective logging accelerates gap-phase regeneration within the degraded forests (*Huang et al.*, 2008).

Such a simplified representation of wood harvest in ESMs has been necessary because models generally do not represent the demographic structure of forests (tree size and stem number distributions) (*Bonan*, 2008). But progress over the past two decades in ecological theory and observations (*Bustamante et al.*, 2015;*Strigul et al.*, 2008;*Hurtt et al.*, 1998;*Moorcroft* et al., 2001) has made it feasible to include vegetation demography more directly into Earth system models through individual to cohort-based vegetation in land models (*Sato et al.*, 2007;*Watanabe et al.*, 2011;*Smith et al.*, 2001;*Smith et al.*, 2014;*Weng et al.*, 2015; *Roy* et al., 2003;*Hurtt et al.*, 1998;*Fisher et al.*, 2015). These vegetation demography modules are relatively new in land models, so tremendous efforts are still under way to improve their parameterizations of resource competition for light, water, and nutrients, recruitment, mortality, and disturbance including both natural and anthropogenic components (*Fisher et al.*, 2017).

In this study, we aim to (1) describe the development of a selective logging module implemented into The Functionally Assembled Terrestrial Ecosystem Simulator (FATES), for simulating anthropogenic disturbances of various intensities to forest ecosystems and their short-term and long-term effects on water, energy, and carbon cycling, and ecosystem dynamics; (2) assess the capability of FATES in simulating site-level water, energy, and carbon budgets, as well as forest structure and composition; (3) benchmark the simulated variables against available observations at the Tapajós National Forest in the Amazon, thus identifying potential directions for model improvement; and (4) assess the recovery trajectory of tropical forest following disturbance under various logging scenarios. In section 2, we provide a brief summary on FATES, introduce the new selective logging module, and describe numerical experiments performed at two sites with data from field survey and flux towers. In section 3, FATES-simulated water, energy,



and carbon fluxes and stocks in intact and disturbed forests are compared to available observations,
and the effects of logging practice and intensity on forest recovery trajectory in terms of carbon
budget, size structure and composition in plant functional types are assessed. Conclusions and
future work are discussed in section 4.

## 2   Model description and study site

### 2.1 The Functionally Assembled Terrestrial Ecosystem Simulator

The Functionally Assembled Terrestrial Ecosystem Simulator (FATES) has been developed as a
numerical terrestrial ecosystem model based on the ecosystem demography representation in the
community land model (CLM), formerly known as CLM (ED) (*Fisher et al.*, 2015). FATES is an
implementation of the cohort-based Ecosystem Demography (ED) concept (*Hurtt et al.*,
1998;*Moorcroft et al.*, 2001) that can be called as a library from an ESM land surface scheme,
currently including CLM (*Oleson et al.*, 2013) or Energy Exascale Earth system model (E3SM)
land model (ELM) (https://climatemodeling.science.energy.gov/projects/energy-exascale-earth-
system-model). In FATES, the landscape is discretized into spatially implicit *patches* each of
which represents land areas with a similar *age since last disturbance*. The discretization of
ecosystems along a disturbance/recovery axis allows the deterministic simulation of successional
dynamics within a typical forest ecosystem. Within each patch, individuals are grouped into
*cohorts* by plant functional types (PFTs) and size classes (SCs), so that cohorts can compete for
light based on their heights and canopy positions. Following disturbance, a patch fission process
splits the original patch into undisturbed and disturbed new patches. A patch fusion mechanism is
implemented to merge patches with similar structures, which helps prevent the number of patches
from growing too big. In addition to the ED concept, FATES also adopted a modified version of
the Perfect Plasticity Approximation (PPA) (*Strigul et al.*, 2008) concept by splitting growing
cohorts between canopy and understory layers as a continuous function of height designed for
increasing the probability of co-existence (*Fisher et al.*, 2010). An earlier version of FATES,
CLM(ED), has been applied regionally to explore the sensitivity of biome boundaries to plant trait
representation (*Fisher et al.*, 2015).
In this study, we specified two plant functional types (PFTs) in FATES corresponding to
early successional and late successional plants, representative of the primary axis of variability in





tropical forests (*Reich* 2014). The early successional PFT is light-demanding, and grows rapidly
under high light conditions common prior to canopy closure. This PFT has low density woody
tissues, shorter leaf and root lifetimes, and a higher background mortality compared to the late
successional PFT that has dense woody tissues, longer leaf and root lifetimes, and lower
background mortality (*Brokaw*, 1985;*Whitmore*, 1998) and thus can survive under deep shade and
grow slowly under closed canopy.

147         The key parameters that differentiate the two PFTs in FATES are listed in Table 1, including

specific leaf area at the canopy top ($SLA_0$), the maximum rate of carboxylation at 25 °C ($V_{cmax25}$),
specific wood density, background mortality, leaf and fine root longevity, and leaf C:N ratio. The
parameter ranges were selected based on literature for tropical forests. Specifically, it has been
reported that SLA values ranges from 0.007-0.039 $m^2$ $gC^{-1}$ (*Wright et al.*, 2004), $V_{cmax25}$ ranges
between 10.1 and 105.7 µmol $m^{-2}$ $s^{-1}$ (*Domingues et al.*, 2005), Specific wood density and
background mortality were set to be 0.5 and 0.9 g $cm^3$ for early and late succession PFTs,
consistent with those used in the Ecosystem Demography Model version 2 for Amazon forests
(Longo et al., in review) . For simplicity, leaf longevity and root longevity were set to be the same
for each PFT (i.e., 0.9 yr and 2.6 yr for early and late successional PFTs) following the range in
*Trumbore and Barbosa De Camargo* (2009).

158         Given that both $SLA_0$ and $V_{cmax25}$ span wide ranges, and have been identified as the most

sensitive parameters in FATES in a previous study (*Massoud et al.*, 2019), we performed one-at-
a-time sensitivity tests by perturbing them within the reported ranges. Based on these tests, it is
evident that these parameters not only affect water, energy, carbon budget simulations, but also
the coexistence of the two PFTs. In the current version of FATES, co-existence of PFTs is not
assured for all parameter combinations, even if they are both within reasonable ranges, on account
of competitive exclusion feedback processes that prevent co-existence in the presence of large
discrepancies in plant growth and reproduction rates (*Fisher et al*. 2010; *Bohn et al*. 2011). In
order to demonstrate FATES' capability in simulating water, energy, carbon budgets as well as
forest structure and composition in a holistic way, we chose to report results based on a set of
parameter values that produces reasonable, stable fractions of two PFTs, as reported in Table 1.



## 2.2 The selective logging module

The new selective logging module in FATES mimics the ecological, biophysical, and biogeochemical processes following a logging event. The module (1) specifies the timing and areal extent of a logging event; (2) calculates the fractions of trees that are damaged by direct felling, collateral damage, and infrastructure damage, and adds these size-specific plant mortality types to FATES; (3) splits the logged patch into disturbed and intact new patches; (4) applies the calculated survivorship to cohorts in the disturbed patch; and (5) transports harvested logs off-site by adding the remaining necromass from damaged trees into coarse woody debris and litter pools.

The logging module structure and parameterization is based on detailed field and remote sensing studies (*Putz et al*., 2008;*Asner et al*., 2004;*Pereira Jr et al*., 2002;*Asner et al*., 2005;*Feldpausch et al*., 2005). Logging infrastructure including roads, skids, trails, and log decks are represented (Figure 1). The construction of log decks used to store logs prior to road transport leads to large canopy openings but their contribution to landscape-level gap dynamics is small. In contrast, the canopy gaps caused by tree felling are small but their coverage is spatially extensive at the landscape scale. Variations in logging practices significantly affect the level of disturbance to tropical forest following logging (*Pereira Jr et al.*, 2002;*Macpherson et al*., 2012;*Dykstra*, 2002;*Putz et al*., 2008). Logging operations in the tropics are often carried out with little planning, and typically use heavy machinery to access the forests accompanied by construction of excessive roads and skid trails, leading to unnecessary tree fall and compaction of the soil. We refer to these typical operations as conventional logging (CL). In contrast, reduced impact logging (RIL) is a practice with extensive pre-harvest planning, where trees are inventoried and mapped out for the most efficient and cost-effective harvest and *seed trees* are deliberately left on site to facilitate faster recovery. Through planning, the construction of skid trails and roads, soil compaction and disturbance can be minimized. Vines connecting trees are cut and tree-fall directions are controlled to reduce damages to surrounding trees. Reduced impact logging results in consistently less disturbance to forests than conventional logging (*Pereira Jr et al.* 2002; *Putz et al.* 2008).

The FATES logging module was designed to represent a range of logging practices in field operations at a landscape level. Once logging events are activated, we define three types of mortality associated with logging practices: direct-felling mortality ($lmort_{direct}$), collateral mortality ($lmort_{collateral}$), and mechanical mortality ($lmort_{mechanical}$). The direct felling mortality represents the fraction of trees selected for harvesting that are greater or equal to a diameter





threshold (this threshold is defined by the diameter at breast height (DBH) = 1.3 m denoted as

DBH$_{min}$); collateral mortality denotes the fraction of adjacent trees that killed by felling of the

harvested trees; and the mechanical mortality represents the fraction of trees killed by construction

of log decks, skid trails and roads for accessing the harvested trees, as well as storing and

transporting logs offsite (Figure 1a). In a logging operation, the loggers typically avoid large trees

when they build log decks, skids, and trails by knocking down relatively small trees as it is not

economical to knock down large trees. Therefore, we implemented another DBH threshold,

DBH$_{max\_infra}$, so that only a fraction of trees ≤ DBH$_{max\_infra}$ (called mechanical damage fraction)

are removed for building infrastructure (*Feldpausch et al*., 2005).

To capture the disturbance mechanisms and degree of damage associated with logging

practices at the landscape level, we apply the mortality types following a workflow designed to

correspond to field operations. In FATES, as illustrated in Figure 2, individual trees of all plant

functional types (PFTs) in one patch are grouped into cohorts of similar-sized trees, whose size

and population sizes evolve in time through processes of recruitment, growth, and mortality. For

the purpose of reporting and visualizing the model state, these cohorts are binned into a set of 13

fixed size classes in terms of the diameter at the breast height (DBH) (i.e., 0 – 5, 5 – 10, 10 – 15,

15 – 20, 20 – 30 , 30 – 40, 40 – 50, 50 – 60, 60 – 70, 70 – 80, 80 – 90, 90 – 100, and ≥100 cm).

Cohorts are further organized into canopy and understory layers, which are subject to different

light conditions (Figure 2a). When logging activities occur, the canopy trees and a portion of big

understory trees lose their crown coverage through direct felling for harvesting logs, or as a result

of collateral and mechanical damages (Figure 2b). The fractions of (only the) canopy trees affected

by the three mortality mechanisms are then summed up to specify the areal percentages of an old

(undisturbed) and a new (disturbed) patch caused by logging in the patch fission process as

discussed section 2.1 (Figure 2c). After patch fission, the canopy layer over the disturbed patch

is removed, while that over the undisturbed patch stays untouched (Figure 2d). In the undisturbed

patch, the survivorship of understory trees is calculated using an understory death fraction

consistent with whose default value corresponds to that used for natural disturbance (i.e., 0.5598).

To differentiate logging from natural disturbance, a slightly elevated, logging-specific understory

death fraction is applied in the disturbed patch instead at the time of the logging event. Based on

data from field surveys over logged forest plots in southern Amazon (*Feldpausch et al*., 2005),

understory death fraction corresponding to logging is now set to be 0.65 as the default, but can be




modified via the FATES parameter file (Figure 2e). Therefore, the logging operations will change
the forest from the undisturbed state shown in Figure 2a to a disturbed state in Figure 2f in the
logging module. It is worth mentioning that the newly generated patches are tracked according to
*age since disturbance* and will be merged with other patches of similar canopy structure following
the patch fusion processes in FATES in later time steps of a simulation, pending the inclusion of
separate land-use fractions for managed and unmanaged forest.
Logging operations affect forest structure and composition, and also carbon cycling (*Palace et
al.*, 2008) by modifying the live biomass pools and flow of necromass (Figure 3). Following a
logging event, the logged trunk products from the harvested trees are transported off-site (as an
added carbon pool for resource management in the model), while their branches enter the coarse
woody debris (CWD) pool, and their leaves and fine roots enter the litter pool. Similarly, trunks
and branches of the dead trees caused by collateral and mechanical damages also become CWD,
while their leaves and fine roots become litter. Specifically, the densities of dead trees as a result
of direct felling, collateral, and mechanical damages in a cohort are calculated as follows:

$$
\begin{aligned}
D_{\text{direct}} &= \text{lmort}_{\text{direct}} \times \frac{n}{A} \\
D_{\text{collateral}} &= \text{lmort}_{\text{collateral}} \times \frac{n}{A} \\
D_{\text{mechanical}} &= \text{lmort}_{\text{mechanical}} \times \frac{n}{A}
\end{aligned}
\tag{1}
$$

where $A$ stands for the area of the patch being logged, and $n$ is the number of individuals in the
cohort where the mortality types apply (i.e., as specified by the size thresholds, $\text{DBH}_{\text{min}}$ and
$\text{DBH}_{\text{max\_infra}}$). For each cohort, we denote $D_{\text{indirect}} = D_{\text{collateral}} + D_{\text{mechanical}}$ and $D_{\text{total}} =$
$D_{\text{direct}} + D_{\text{indirect}}$, respectively.
Leaf litter ($\text{Litter}_{\text{leaf}}$, [kg C]) and root litter ($\text{Litter}_{\text{root}}$, [kg C]) at the cohort level are then
calculated as:

$$
\text{Litter}_{\text{leaf}} = D_{\text{total}} \times B_{leaf} \times A
\tag{2}
$$

$$
\text{Litter}_{\text{root}} = D_{\text{total}} \times (B_{root} + B_{store}) \times A
\tag{3}
$$

where $B_{leaf}$, $B_{root}$, and $B_{store}$ are live biomass in leaves and fine roots, and stored biomass in
the labile carbon reserve in all individual trees in the cohort of interest.
Following the existing CWD structure in FATES (*Fisher et al.*, 2015), CWD in the logging
module is first separated into two categories: above-ground CWD and below-ground CWD.



Within each category, four size classes are tracked based on their source, following Thonicke et
al. (2010): trunks, large branches, small branches and twigs. Above-ground CWD from trunks
($CWD_{trunk\_agb}$, [kg C]) and large branches/small branches/twig ($CWD_{branch\_agb}$, [kg C]) are
calculated as follows:

$$CWD_{trunk\_agb} = D_{indirect} \times B_{stem\_agb} \times f_{trunk} \times A \qquad (4)$$

$$CWD_{branch\_agb} = D_{total} \times B_{stem\_agb} \times f_{branch} \times A \qquad (5)$$

where $B_{stem\_agb}$ is the amount of above ground stem biomass in the cohort, $f_{trunk}$ and $f_{branch}$
represent the fraction of trunks and large branches/small branches/twig. Similarly, the below-
ground CWD from trunks ($CWD_{trunk\_bg}$, [kg C]) and branches/twig ($CWD_{branch\_bg}$, [kg C]) are
calculated as follows:

$$CWD_{trunk\_bg} = D_{total} \times B_{root\_bg} \times f_{trunk} \times A \qquad (6)$$

$$CWD_{branch\_bg} = D_{total} \times B_{root\_bg} \times f_{branch} \times A \qquad (7)$$

where $B_{croot}$ [kg C] is the amount of coarse root biomass in the cohort. Site-level total litter and
CWD inputs can then be obtained by integrating the corresponding pools over all the cohorts in
the site. To ensure mass conservation,

$$\Delta B = \Delta Litter + \Delta CWD + trunk\_product \qquad (8)$$

where ΔB is total loss of biomass due to logging, Δlitter and ΔCWD are the increments in litter
and CWD pools, and *trunk_product* represents harvested logs shipped offsite.
Following the logging event, the forest structure and composition in terms of cohort
distributions, as well as the live biomass and necromass pools are updated. Following this logging
event update to forest structure, the native processes simulating physiology, growth and
competition for resources in and between cohorts resume. Since the canopy layer is removed in
the disturbed patch, the existing understory trees are promoted to the canopy layer, but, in general,
the canopy is incompletely filled in by these newly-promoted trees, and thus the canopy does not
fully close. Therefore, more light can penetrate and reach the understory layer in the disturbed
patch, leading to increases in light-demanding species in the early stage of regeneration, followed
by a succession process in which shade tolerant species dominate gradually.





### 2.3 Study site and data

In this study, we used data from two evergreen tropical forest sites located in the Tapajós National Forest (TNF), Brazil (Figure 1b). These sites were established during the Large-Scale Biosphere-Atmosphere Experiment in Amazonia (LBA), and are selected because of data availability including those from forest plot surveys and two flux towers established during the LBA period (*Keller et al.*, 2004a). These sites were named after distances along the BR-163 highway from Santarém: km67 (54°58'W, 2°51'S) and km83 (54°56'W, 3°3'S). They are situated on a flat plateau and were established as a control-treatment pair for a selective logging experiment. Tree felling operations were initiated at km83 in September 2001 for a period of about two months. Both sites are similar with mean annual precipitation of ~2000 mm, and mean annual temperature of 25 °C, on nutrient-poor clay oxisols with low organic content (*Silver et al.*, 2000).

Prior to logging, both sites were old-growth forests with limited previous human disturbances caused by hunting, gathering Brazil nuts, and similar activities. A comprehensive set of meteorological variables, as well as land-atmosphere exchanges of water, energy, and carbon fluxes have been measured by an eddy covariance tower at a hourly time step over the period of 2002 to 2011, including precipitation, air temperature, surface pressure, relative humidity, incoming shortwave and longwave radiation, latent and sensible heat fluxes, and net ecosystem exchange (NEE) (Hayek et al., 2018). Another flux tower was established at km83, the logged site, with hourly meteorological and eddy covariance measurements in the period of 2000-2003 (*Miller et al.*, 2004;*Goulden et al.*, 2004;*Saleska et al.*, 2003). The towers are listed as BR-Sa1 and BR-Sa3 in the AmeriFlux network (https://ameriflux.lbl.gov).

These tower and biometric based observations were summarized to quantify logging-induced perturbations on old-growth Amazonian forests in *Miller et al.* (2011) and are used in this study to benchmark the model simulated carbon budget. Over the period of 1999 to 2001, all trees ≥ 35cm in DBH in 20 ha of forest in four 1-km long transects within the km67 footprint were inventoried, as well as trees ≥ 10 cm in DBH on subplots with an area of ~4 ha. At km83, inventory surveys on trees ≥ 55 cm in DBH were conducted in 1984 and 2000, and another survey on trees > 10 cm in DBH was conducted in 2000 (*Miller et al.*, 2004). Estimates of above ground biomass (AGB) were then derived using allometric equation for Amazon forests (*Rice et al.*, 2004;*Chambers* et al., 2004;*Keller* et al., 2001). Necromass (≥2 cm diameter) production was also measured approximately every six months in a 4.5-year period from November 2001 through February 2006



in logged and undisturbed forest at km83 (*Palace et al*., 2008). Field measurements of ground
disturbance in terms of number of felled trees, areas disturbed by collateral and mechanical
damages were also conducted at a similar site in Pará state along multitemporal sequences of post-
harvest regrowth of 0.5–3.5 yr (*Asner et al*., 2004;*Pereira Jr et al*., 2002).
Table 2 provides a summary of stem density and basal area distribution across size classes at
km83 based on the biomass survey data (*Menton et al.* 2011; *de Sousa et al.*, 2011). To facilitate
comparisons with simulations from FATES, we divided the inventory into early and late
succession PFTs using threshold of 0.7 g cm$^{-3}$ for specific wood density, consistent with the
definition of these PFTs in Table 1. As shown in Table 2, prior to the logging event in year 2000,
this forest was composed of 399, 30 & 30 trees per hectare in size classes of 10-30 cm, 30-50 cm,
and ≥50 cm respectively; Following logging, the numbers were reduced to 396, 29, and 18 trees
per hectare, losing ~1.3% of trees ≥10 cm in size. The changes in stem density (SD) were caused
by different mechanisms for different size classes. The reduction in stem density of 2 ha$^{-1}$ in the
≥50 cm size class was caused by timber harvest directly, while the reductions of 3 ha$^{-1}$ and 1 ha$^{-1}$
in the 10-30 cm and 30-50 cm size classes were caused by collateral and mechanical damages.
Corresponding to the loss of trees in logging operations, basal area (BA) decreased from 3.9, 4.0,
and 12.9 m$^2$ ha$^{-1}$ to 3.8, 3.9, and 10.8 m$^2$ ha$^{-1}$, and above ground biomass (AGB) decreased from
3.8, 2.3, and 10.4 kg C m$^{-2}$ to 3.8, 2.2, 8.7 kg C m$^{-2}$ in the 10-30 cm, 30-50 cm, and ≥50 cm size
class, respectively.

### 2.4 Numerical Experiments

In this study, the gap-filled meteorological forcing data for Tapajós National Forest processed by
*Longo* (2014)  are used to drive the CLM(FATES) model.  Characteristics of the sites, including
soil texture, vegetation cover fraction, and canopy height, were obtained from the LBA-Data
Model Intercomparison Project (*de Gonçalves et al*., 2013). Specifically, soil at km 67 contains
90% clay and 2% sand, while soil at km 83 contains 80% clay and 18% sand. Both sites are covered
by tropical evergreen forest at ~ 98% within their footprints, with the remaining 2% assumed to
be covered by bare soil. As discussed in *Longo et al*. (2018), who deployed the Ecosystem
Demography model version 2 at this site, soil texture and hence soil hydraulic parameters are
highly variable even with the footprint of the same eddy covariance tower, and could have
significant impacts on not only water and energy simulations, but also simulated forest





composition and carbon stocks and fluxes. Further, generic pedo-transfer functions designed to capture temperate soils typically perform poorly in clay-rich Amazonian soils (*Fisher et al.* 2008, *Tomasella and Hodnett*, 1998). Because we focus on introducing the FATES-logging, we leave for forthcoming studies the exploration of the sensitivity of the simulations to soil texture and other critical environmental factors.

CLM(FATES) was initialized using soil texture at km83 (i.e., 80% clay and 18% sand) from bare ground and spun up for 800 years until the carbon pools and forest structure (i.e., size distribution) and composition of PFTs reached equilibrium, by recycling the meteorological forcing at km67 (2001-2011) as the sites are close enough. The final states from spin-up were saved as the initial condition for follow-up simulations. An *intact* experiment was conducted by running the model over a period of 2001 to 2100 without logging by recycling the 2001-2011 forcing using the parameter set in Table 1. The atmospheric $CO_2$ concentration was assumed to be a constant of 367 ppm over the entire simulation period, consistent with the $CO_2$ levels during the logging treatment (*Dlugokencky et al.*, 2017).

We specified an experimental logging event in FATES on 1 September 2001 (Table 3). It was reported by *Figueira et al.* (2008) that following the reduced impact logging event in September 2001, 9% of the trees greater or equal to $DBH_{min} = 50$ cm were harvested, with an associated collateral damage fraction of 0.009 for trees $\geq DBH_{min}$. $DBH_{max\_infra}$ is set to be 30 cm, so that only a fraction of trees $\leq$ 30 cm are removed for building infrastructure (*Feldpausch et al.*, 2005). This experiment is denoted as the $RIL_{low}$ experiment in Table 2 and is the one that matches the actual logging practice at km83.

We recognize that the harvest intensity in September 2001 at km83 was extremely low. Therefore, in order to study the impacts of different logging practices and harvest intensities, three additional logging experiments were conducted as listed in Table 3: conventional logging with high intensity ($CL_{high}$), conventional logging with low intensity ($CL_{low}$), and reduced impact logging with high intensity ($RIL_{high}$). The high intensity logging doubled the direct felling fraction in $RIL_{low}$ and $CL_{low}$, as shown in the $RIL_{high}$ and $CL_{high}$ experiments. Compared to the RIL experiments, the CL experiments feature elevated collateral and mechanical damages as one would observe in such operations. All logging experiments were initialized from the spun-up state using site characteristics at km83 previously discussed and were conducted over the period of 2001-2100 by recycling meteorological forcing from 2001- 2011.





## 3 Results and discussions

### 3.1 Simulated energy and water fluxes

Simulated monthly mean energy and water fluxes at the two sites are shown and compared to available observations in Figure 4. The performances of the simulations closest to site conditions were compared to observations and summarized in Table 4 (i.e., intact for km67 and RIL$_{low}$ for km83). The observed fluxes as well as their uncertainty ranges noted as Obs67 and Obs83 from the towers were obtained from *Saleska et al.* (2013), consistent with those in *Miller et al*. (2011). As shown in Table 4, the simulated mean (±standard deviation) latent heat (LH), sensible heat (SH), and net radiation (Rn) fluxes at km83 in RIL$_{low}$ over the period of 2001-2003 are 108.3± 20.8, 20.5 ± 24.3 and 128.9 ± 15.5 W m$^{-2}$, compared to tower-based observations of 101.6 ± 8.0, 25.6 ± 5.2 and 129.3 ± 18.5 W m$^{-2}$. Therefore, the simulated and observed Bowen ratios are 0.16 and 0.20 at km83, respectively. This result suggests that at an annual time step, the observed partitioning between LH and SH are reasonable. However, at seasonal scales, even though net radiation is captured by CLM (FATES), the model does not adequately partition sensible and latent heat fluxes. This is particularly true for sensible heat fluxes as the model simulates large seasonal variabilities in SH when compared to observations at the site (i.e., standard deviations of monthly-mean simulated SH are ~ 24.3 W m$^{-2}$, while observations are ~ 5.2 W m$^{-2}$). As illustrated in figures 4(c) and 4(d), the model significantly overestimates SH in the dry season (June-December), while it slightly underestimates SH in the wet season. It is worth mentioning that incomplete closure of the energy budget is common at eddy covariance towers (*Wilson et al.*, 2002;*Foken*, 2008) and has been reported to be ~87% at the two sites (*Saleska et al*., 2003). Nevertheless, some of the mismatches between observations and simulations can be attributed to structural problems in this version of FATES. For example, the mean simulated leaf area indices (LAIs) are ~2.4 m$^2$m$^{-2}$, while observations suggest that LAIs at these sites ranges from 5-7 m$^2$m$^{-2}$ (*Doughty and Goulden*, 2008;*Brando et al*., 2010). The low LAI bias in the model leads to lower simulated LH, and in turn the overestimation of SH to conserve energy.

Figure 4(j) shows the comparison between simulated and observed (*Goulden et al*., 2010) volumetric soil moisture content (m$^3$m$^{-3}$) at top 10 cm. This comparison reveals another model structural deficiency, that is, even though the model simulates higher soil moisture contents compared to observations (a feature generally attributable to the soil moisture retention curve), the transpiration beta factor, the down-regulating factor of transpiration from plants, fluctuates



significantly over a wide range, and can be as low as 0.13 in the dry season. In reality flux towers
in the Amazon generally do not show severe moisture limitations in the dry season (*Fisher et al*.
2007). The lack of limitation is typically attributed to the plant's ability to extract soil moisture
from deep soil layers, a phenomenon that is difficult to simulate using a classical beta function
(*Baker et al*. 2008), and potentially is reconcilable using hydrodynamic representation of plant
water uptake (*Powell et al*. 2014; *Christoffersen et al*. 2016) as are in the final stages of
incorporation into the FATES model. Consequently, the model simulates consistently low ET
during dry seasons (figures 4(e) and 4(f)), while observations indicate that canopies are highly
productive owing to adequate water supply to support transpiration and photosynthesis, which
could further stimulate coordinated leaf growth with senescence during the dry season (*Wu et al*.

420  2016; 2017).


### 3.2 Carbon budget, and forest structure and composition in the intact forest

Figures 5, 6, and 7 show simulated carbon pools and fluxes, which are tabulated in Table 5 as well.
As shown in Figure 5, prior to logging, the simulated above ground biomass and necromass (CWD
+ litter) are 155 Mg C ha$^{-1}$ and 41.1 Mg C ha$^{-1}$, compared to 165 Mg C ha$^{-1}$ and 58.4 Mg C ha$^{-}$
$^{1}$ based on permanent plot measurements. The simulated carbon pools are generally lower than
observations reported in *Miller et al.* (2011) but are within reasonable ranges, as errors associated
with these estimates could be as high as 50% due to issues related to sampling and allometric
equations, as discussed in *Keller et* al. (2001). The lower biomass estimates are consistent with the
finding of excessive soil moisture stress during the dry season, and low LAI in the model.
Combining forest inventory and eddy covariance measurements, *Miller et al.* (2011) also
provides estimates for net ecosystem exchange (NEE), gross primary production (GPP), net
primary production (NPP), ecosystem respiration (ER), heterotrophic respiration (HR), and
autotrophic respiration (AR). As shown in Table 5, the model simulates a NPP of 8.9 Mg C ha$^{-2}$
yr$^{-1}$ and a HR of 9.4 Mg C ha$^{-2}$ yr$^{-1}$, in comparison to the estimated NPP of 9.5 Mg C ha$^{-2}$ yr$^{-1}$ and
HR of 8.9 Mg C ha$^{-2}$ yr$^{-1}$ in the intact forest based on field measurements. This suggests that despite
the low LAI, the model nonetheless captures the turnover of the live carbon pools and the decay
rates of the necromass pools reasonably well. However, the model simulates much lower values
in GPP (17.6 Mg C ha$^{-2}$ yr$^{-1}$), AR (8.7 Mg C ha$^{-2}$ yr$^{-1}$), and ER (18.1 Mg C ha$^{-2}$ yr$^{-1}$), when
compared to values estimated from the observations (32.6 Mg C ha$^{-2}$ yr$^{-1}$ for GPP, 23.1 Mg C ha$^{-}$



$^2$ yr$^{-1}$ for AR, and 31.9 Mg C ha$^{-2}$ yr$^{-1}$ for ER).  The low biases in simulated AGB, GPP, AR and
leaf area index (figures 4g and 4h) suggests that this version of the model suffers from parametric
uncertainties in its capability of establishing enough live plant tissues for photosynthesis and
autotropic respiration at the patch level that are the subject of ongoing updates and modifications.
Compensating errors in the gross fluxes, however, produce reasonable NPP estimates, making all
the ecosystem processes downstream of NPP within the observed ranges.

447       Consistent with the carbon budget terms, Table 5 lists the simulated and observed values of

stem density (ha$^{-1}$) in different size classes in term of DBH. The model simulates 232 trees per
hectare with DBHs greater than or equal to 10 cm in the intact forest, compared to 459 trees per
hectare from observed inventory. In terms of distribution across the DBH classes of 10-30 cm, 30-
50 cm, and ≥50 cm, 145, 43, and 44 N ha$^{-1}$ of trees were simulated, while 399, 30, and 30 N ha$^{-1}$
were observed in the intact forest. In general, this version of FATES is simulating a less dense
forest, with a forest structure biased toward larger trees, a feature that may result from allometric
considerations. Trees have a maximum crown area in FATES, after which DBH increases but
spatial extent does not. If this crown-area threshold is too high, a limited number of crowns will
fit into the canopy, leading to low biases in number density. In addition to size distribution, by
parametrizing early and late successional PFTs (Table 1), FATES is capable of simulating the co-
existence of the two PFTs, therefore the PFT-specific trajectories of stem density, basal area,
canopy and understory mortality rates. We will discuss these in section 3.4.


## 3.3 Effects of logging on water, energy, and carbon budgets

The response of energy and water budgets to different levels of logging disturbances are illustrated
in Table 4 and Figure 4. Following the logging event, the LAI is reduced proportionally to the
logging intensities (-7%, -15%, -9% and -17% for RL$_{low}$, RL$_{high}$, CL$_{low}$, and CL$_{high}$ respectively
in September 2001, see figure 4h).  Leaf area index recovers within three years to its pre-logging
level, or even to slightly higher levels as a result of the improved light environment following
logging leading to changes in forest structure and composition (to be discussed in section 3.4). In
response to the changes in stem density and LAI, discernible differences are found in all energy
budget terms. For example, less leaf area leads to reductions in LH (-0.5%, -1.0%, -6.9%, -7.4%)
and increases in SH (4.0%, 8.0%, 4.4%, and 8.6%) proportional to the damage levels (i.e., RL$_{low}$,




$RL_{high}$, $CL_{low}$, and $CL_{high}$) in the first three years following the logging event when compared to
the control simulation. Energy budget responses scale with the level of damage, so that the biggest
differences are detected in the $CL_{high}$ scenario, followed by $RIL_{high}$, $CL_{low}$ and $RIL_{low}$. The
difference in simulated water and energy fluxes between the $RIL_{low}$ (i.e., the scenario that is the
closest to the experimental logging event) and intact cases is the smallest, as the level of damage
is the lowest among all scenarios.

478       As with LAI, the water and energy fluxes recover rapidly in 3-4 years following logging.

*Miller et al.* (2011) compared observed sensible and latent heat fluxes between the control (km67)
and logged sites (km83). They found that in the first three years following logging, the between-
sites difference (i.e., logged – control) in LH reduced from $19.7 \pm 2.4$ to $15.7 \pm 1.0$ W m$^2$, and that
in SH increased from $3.6 \pm 1.1$ to $5.4 \pm 0.4$ W m$^2$. When normalized by observed fluxes during the
same periods at km83, these changes correspond to a -4% reduction in LH and a 7% increase in
SH, compared to the -0.5% and 4% differences in LH and SH between $RL_{low}$ and the control
simulations. In general, both observations and our modelling results suggest that the impacts of
reduced impact logging on energy fluxes are modest and that the energy and water fluxes can
quickly recover to their pre-logging conditions at the site.

488       Figures 6 and 7 show the impact of logging on carbon fluxes and pools at a monthly time

step, and the corresponding annual fluxes and changes in carbon pools are summarized in Table 5.
The logging disturbance leads to reductions in GPP, NPP, AR, and AGB, and increases in ER,
NEE, HR, and CWD.  The impacts of logging on the carbon budgets are also proportional to
logging damage levels. Specifically, logging reduces the simulated AGB from 155 Mg C ha$^{-1}$
(intact) to 138.0 Mg C ha$^{-1}$ ($RIL_{low}$), 119.3 Mg C ha$^{-1}$ ($RIL_{high}$), 137.8 Mg C ha$^{-1}$ ($CL_{low}$) and 118.9
($CL_{high}$), while increases the simulated necromass pool (CWD + litter) from 41.1 Mg C ha$^{-1}$ in the
intact case to 59.6 Mg C ha$^{-1}$ ($RIL_{low}$), 79.5 Mg C ha$^{-1}$ ($RIL_{high}$), 60.0 Mg C ha$^{-1}$ ($CL_{low}$) and 80.1
($CL_{high}$). For the case closest to the experimental logging event ($RIL_{low}$), the changes in AGB and
necromass from the intact case are -17 Mg C ha$^{-1}$ (11%) and 18.5 Mg C ha$^{-1}$ (45%), in comparison
to observed changes of -22 Mg C ha$^{-1}$ in AGB (12%) and 16 Mg C ha$^{-1}$ (27%) in necromass from
*Miller et al.* (2011), respectively.  The negative model biases in carbon pools, GPP, ER, and AR
(see section 3.2) propagate into their estimates following disturbance (Table 5), but the directions
of their changes are reasonable when compared to observations (i.e., decreases in GPP, ER, and
AR following logging).  On the other hand, the simulations indicate that the forest could be turned





from a small carbon source (0.5 Mg C ha$^{-1}$ yr$^{-1}$) to a larger carbon source in 1-5 years following
logging, while observations from the tower suggested that the forest was a carbon sink or a modest
carbon source (-0.6 ± 0.8 Mg C ha$^{-1}$ yr$^{-1}$) prior to logging, and turned into a carbon sink in three
years following logging. Such a mismatch between observations and simulations is a result of a
less productive forest in the model.

508       The recovery trajectories following logging are also shown in figures 6, 7, and Table 5. It

takes more than 70 years for AGB to return to its pre-logging levels, but the recovery of carbon
fluxes such as GPP, NPP, and AR is much faster (i.e., within five years following logging). The
initial recovery rates of AGB following logging are faster for high-intensity logging because
increased light reaching the forest floor, as indicated by the steeper slopes corresponding to the
CL$_{high}$ and RIL$_{high}$ scenarios compared to those of CL$_{low}$ and RIL$_{low}$ (figure 9h). While this finding
is consistent with previous observational and modelling studies (*Mazzei et* al., 2010; *Huang and*
*Asner*, 2010) in that the damage level determines the number of years required to recover the
original AGB, and the AGB accumulation rates in recently logged forests are higher than that in
intact forest, the simulated recovery time is slower than that reported in literature. For example, by
synthesizing data from 79 permanent plots at 10 sites across the Amazon basin, *Ruttishauser et al.*
(2016) and *Piponiot et al.* (2018) show that it requires 12, 43, and 75 years for the forest to recover
with initial losses of 10, 25, or 50% in AGB. The slow recovery time in the simulation might be
attributed to the low GPP bias in this version of CLM (FATES).    Corresponding to the changes
in AGB, logging introduces a large amount of necromass to the forest floor, with the highest
increases in the CL$_{high}$ and RIL$_{high}$ scenarios. As shown in Figure 7(d) and Table 5, necromass and
CWD pools return to the pre-logging level in ~15 years. Meanwhile, HR in RIL$_{low}$ stays elevated
in five years following logging but converges to that from the intact simulation in ~10 years, which
is consistent with observation (*Miller et al. 2011*; Table 5).

**3.4 Effects of logging on forest structure and composition**

The capability of the CLM(FATES) model to simulate vegetation demographics, forest structure
and composition, while simulating the water, energy, and carbon budgets simultaneously (Fisher
et al. 2017) allows interrogation of the modelled impacts of alternative logging practices on forest
size structure. Table 6 shows forest structure in terms of stem density distribution across size
classes from the simulations compared to observations from the site, while figures 8 and 9 further




break it down into early and late succession PFTs and size classes in terms of stem density and
basal areas. As discussed in section 2.2 and summarized in Table 3, the logging practices, reduced
impact logging and conventional logging, differ in terms of pre-harvest planning and actual field
operation to minimize collateral and mechanical damages, while the logging intensities (i.e., high
and low) indicate the target direct felling fractions. The corresponding outcomes of changes in
forest structure in comparison to the intact forest, as simulated by FATES, are summarized in
tables 6 and 7. The conventional logging scenarios (i.e., $CL_{high}$ and $CL_{low}$), feature more losses in
small trees less than 30 cm in DBH, when compared to the smaller reduction in stem density in
size classes less than 30 cm in DBH in the reduced impact logging scenarios (i.e., $RIL_{high}$ and
$RIL_{low}$). Scenarios with different logging intensities (i.e., high and low) result in different direct
felling intensity. That is, the number of surviving large trees (DBH ≥ 30 cm) in $RIL_{low}$ and $CL_{low}$
is 81 ha$^{-1}$, but those in $RIL_{high}$ and $CL_{high}$ is 75 ha$^{-1}$.
In response to the improved light environment after removal of large trees, early successional
trees quickly establish and populate the tree fall gaps following logging in 2-3 years as shown
Figure 8a). Stem density in the <10 cm size classes is proportional to the damage levels (i.e.,
ranked as $CL_{high} > RIL_{high} > CL_{low} > RIL_{low}$), followed by a transition to late successional trees in
later years when the canopy is closed again (Figure 8b). Such a successional process is also evident
in figures 9(a) and 9(b) in terms of basal areas. The number of early successional trees then slowly
declines afterwards but is sustained throughout the simulation as a result of natural disturbances.
Such a shift in the plant community towards light-demanding species following disturbances is
consistent with observations reported in literature (*Baraloto et al.*, 2012; *Both et al.*, 2018).
Following regeneration in logging gaps, a fraction of the late successional trees wins the
competition within the 0-10 cm size classes and is promoted to the 10-30 cm size classes in about
10 years following the disturbances (figures 8d and 9d). Then a fraction of those trees
subsequently enter the 30-50 cm size classes in 20-40 years following the disturbance (figures 8f
and 9f) and so on through larger size classes afterwards (figures 8h and 9h). We note that despite
the goal of achieving a deterministic and smooth averaging across discrete stochastic disturbance
events using the ecosystem demography approach (*Moorcroft et al.*, 2001) in FATES, the
successional process described above, as well as the total numbers of stems in each size bin, shows
evidence of episodic and discrete waves of population change. These arise due to the required




discretization of the continuous time-since-disturbance heterogeneity into patches, combined with
the current maximum cap on the number of patches in FATES (10 per site).
As discussed in section 2.4, the understory early successional trees have a high mortality
(figure 10a) compared to the mortality (figure 10b) of understory late successional trees because
they are shade intolerant. As a result, early successional trees can barely survive in the understory.
Therefore, mortality for understory early successional trees cannot be calculated due to the lack of
population (figures 10c, e, and g). The mortality of large late successional understory trees
gradually increases as more light and water are needed to sustain the trees as they grow larger
(figures 10d, f), and drops again due to lack of population in the >50cm size class. The mortality
rates of small canopy trees (both early and late, as shown in figures 11a and b) decline in the first
few years following logging, and then fluctuate at an equilibrium level because only small
disturbed patches can be created as a result of natural disturbances after the initial logging event.
Mortality rates of large canopy trees (figures 11c-h) are pretty stable, indicating that canopy trees
are not light-limited or water-stressed. Basal area is generally higher in late successional PFT than
in early successional PFT (figure 9) despite its high stem density.
**4 Conclusion and Discussions**
In this study, we developed a selective logging module in FATES and parameterized the model to
simulate different logging practices (conventional and reduced impact) with various intensities.
This newly developed selective logging module is capable of mimicking the ecological,
biophysical, and biogeochemical processes at a landscape level following a logging event in a
lumped way by (1) specifying the timing and areal extent of a logging event; (2) calculating the
fractions of trees that are damaged by direct felling, collateral damage, and infrastructure damage,
and adding these size-specific plant mortality types to FATES ; (3) splitting the logged patch into
disturbed and intact new patches; (4) applying the calculated survivorship to cohorts in the
disturbed patch; and (5) transporting harvested logs off-site and adding the remaining necromass
from damaged trees into coarse woody debris and litter pools.
We then applied FATES coupled to CLM to the Tapajós National Forest by conducting
numerical experiments driven by observed meteorological forcing, and benchmarked the
simulations against long-term ecological and eddy covariance measurements. We demonstrated
that the model is capable of simulating site-level water, energy, and carbon budgets, as well as



forest structure and composition holistically, with responses consistent with those documented in the existing literature as follows:

1. The model captures perturbations on energy and water budget terms in response to different levels of logging disturbances. Our modelling results suggest that logging leads to reductions in canopy interception, canopy evaporation and transpiration, as well as elevated soil temperature and soil heat fluxes in magnitudes proportional to the damage levels.

2. The logging disturbance leads to reductions in GPP, NPP, AR, and AGB, and increases in ER, NEE, HR, and CWD. The initial impacts of logging on the carbon budget are also proportional to damage levels as results of different logging practices.

3. Following the logging event, simulated carbon fluxes such as GPP, NPP, and AR recover within five years, but it takes decades for AGB to return to its pre-logging levels. Consistent with existing observational based literature, initial recovery of AGB is faster when the logging intensity is higher in response to improved light environment in the forest but the time to full AGB recovery in higher intensity logging is longer.

4. Consistent with observations at Tapajós, the prescribed logging event introduces a large amount of necromass to the forest floor proportional to the damage level of the logging event, which returns to pre-logging level in ~15 years. Simulated HR in low-damage reduced impact logging scenario stays elevated in five years following logging and declines to be the same as the intact forest in ~10 years.

5. The impacts of alternative logging practices on forest structure and composition were assessed by parameterizing cohort-specific mortality corresponding to direct felling, collateral damage, mechanical damage in the logging module to represent different logging practices (i.e., conventional logging and reduced impact logging) and intensity (i.e., high and low). In all scenarios, the improved light environment after removal of large trees facilitates establishment and growth of early successional trees in the 0-10 cm DBH size class proportional to the damage levels in the first 2-3 year. Thereafter there is a transition to late successional trees in later years when the canopy is closed. The number of early successional trees then slowly declines but is sustained throughout the simulation as a result of natural disturbances.

Given that the representation of gas exchange processes is related to, but also somewhat independent of the representation of ecosystem demography, FATES shows great potential in its capability to capturing ecosystem successional processes in terms of gap-phase regeneration,





competition among light-demanding and shade-tolerant species following disturbance, as well as
responses of energy, water, and carbon budget components to disturbances. The model projections
suggest that while most degraded forests rapidly recover energy fluxes, the recovery times for
carbon stocks, forest size structure and forest composition are much longer. The recovery
trajectories are highly dependent on logging intensity and practices, the difference between which
can be directly simulated by the model. Consistent with field studies, we find through numerical
experiments that reduced impact logging leads to more rapid recovery of the water, energy, and
carbon cycles, allowing forest structure and composition to recover to their pre-logging levels in a
shorter time frame.

## 5  Future work

Currently, the selective logging module can only simulate single logging events.  For regional-
scale applications, it will be crucial to represent forest degradation as a result of logging, fire, and
fragmentation and their combinations that could repeat over a period. Therefore, we will enable
structural changes in FATES to track disturbance histories associated with fire, logging, and
transitions among land use types. Nevertheless, this study lays the foundation to simulate land use
change and forest degradation in FATES, leading the way to direct representation of forest
management practices and regeneration in Earth System Models.
We also acknowledge that as a model development study, we applied the model to a site using
a single set of parameter values and therefore we ignored the uncertainty associated with model
parameters.  We are also working on fixing the low LAI bias in the model. Preliminary testing
suggests that by reducing the penalty for establishing leaf biomass, the low LAI bias could be
significantly mitigated. This improvement will be evaluated in our follow-up studies.
In addition to the low LAI bias, it is clear that down-regulation factor to transpiration, the
beta factor, is very low in the simulations, leading to underestimation of evapotranspiration and
overestimation of sensible heat fluxes in the dry season. On-going efforts in developing more
mechanistic plant hydraulic models (thereby eliminating the need for a beta factor) could
potentially alleviate the problem (*Christofferson et al.* 2016) and will also be reported separately.





**Author contribution**

M.H., M.K., and M. L. conceived the study, conceptualized the design of the logging module, and designed the numerical experiments and analysis. Y. X., M. H., and R. K. coded the module. Y. X., R. K., C. K., R. F., M. H. integrated the module into FATES. M. H. performed the numerical experiments and wrote the manuscript with inputs from all coauthors.

**Acknowledgements**

This research was supported by The Next-Generation Ecosystem Experiments – Tropics project through the Terrestrial Ecosystem Science (TES) program within US Department of Energy's Office of Biological and Environmental Research (BER). RF acknowledges the National Science Foundation via their support of the National Center for Atmospheric Research.  M.L. was supported by the São Paulo State Research Foundation (FAPESP, grant 2015/07227-6).

**Code and data availability**

FATES-CLM has two separate repositories for FATES and CLM at:

https://github.com/NGEET/fates/releases/tag/sci.1.6.0_api.3.0.0

https://github.com/NGEET/fates-clm/releases.

Site information and data at km67 and km83 can be found at http://sites.fluxdata.org/BR-Sa1 and http://sites.fluxdata.org/BR-Sa13..

A README guide to run the model and formatted datasets used to drive model in this study will be made available from the open-source repository XXXXX upon acceptance of the manuscript.

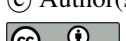



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



**Tables and Figures**
Table 1. FATES Parameters that define early and late successional PFTs

| Parameter names | Units | Early successional PFT | Late successional PFT |
|---|---|---|---|
| Specific leaf area | $m^2\ gC^{-1}$ | 0.016 | 0.015 |
| $V_{cmax}$ at 25°C | $\mu mol\ m^{-2}\ s^{-1}$ | 68 | 60 |
| Specific wood density | $g\ cm^{-3}$ | 0.5 | 0.9 |
| Leaf longevity | yr | 0.9 | 2.6 |
| Background mortality rate | $yr^{-1}$ | 0.035 | 0.014 |
| Leaf C:N | $gC\ gN^{-1}$ | 20 | 40 |
| root longevity | yr | 0.9 | 2.6 |



Table 2. Distributions of stem density (N ha⁻¹), basal area (m2 ha⁻¹) and above ground biomass (Kg C m⁻²)
before and after logging at km83, separated by diameter of breast height (normal text) and aggregated across
all sizes (bold text).

| Time | Before logging | | | After Logging | | |
|---|---|---|---|---|---|---|
| Variables | Early | Late | **Total** | Early | Late | **Total** |
| **Stem Density (N ha⁻¹)** | **264** | **195** | **459** | **260** | **191** | **443** |
| Stem Density (10-30 cm, N ha⁻¹) | 230 | 169 | **399** | 229 | 167 | **396** |
| Stem Density (30-50 cm, N ha⁻¹) | 18 | 12 | **30** | 17 | 12 | **29** |
| Stem Density (≥50 cm, N ha⁻¹) | 16 | 14 | **30** | 14 | 12 | **18** |
| **Basal Area (m² ha⁻¹)** | **11.6** | **9.2** | **21.0** | **10.3** | **8.3** | **18.5** |
| Basal Area (10-30 cm, m² ha⁻¹) | 2.2 | 1.7 | **4.2** | 2.2 | 1.7 | **3.8** |
| Basal Area (30-50 cm, m² ha⁻¹) | 2.4 | 1.6 | **4.2** | 2.4 | 1.6 | **3.9** |
| Basal Area (>=50 cm, m² ha⁻¹) | 7.0 | 5.9 | **12.6** | 5.8 | 5.1 | **10.8** |
| **AGB (Kg C m⁻²)** | **7.6** | **8.9** | **16.5** | **6.8** | **7.9** | **14.7** |
| AGB (10-30 cm, Kg C m⁻²) | 1.8 | 2.0 | **3.8** | 1.8 | 2.0 | **3.8** |
| AGB (30-50 cm, Kg C m⁻²) | 1.1 | 1.1 | **2.3** | 1.1 | 1.1 | **2.2** |
| AGB ((>=50 cm, Kg C m⁻²) | 4.6 | 5.8 | **10.4** | 3.8 | 4.9 | **8.7** |

* based on inventory during the LBA period (*Menton et al.*, 2011; *de Sousa et al.*, 2011)





Table 3. Cohort-level fractional damage fractions in different logging scenarios

| Scenarios | Conventional Logging | | Reduced Impact Logging | |
|---|---|---|---|---|
| | High | Low | High (KM83×2) | Low (KM83) |
| **Experiments** | $CL_{high}$ | $CL_{low}$ | $RIL_{high}$ | $RIL_{low}$ |
| Direct felling fraction (DBH ≥ $DBH_{min}$[1]) | 0.18 | 0.09 | 0.24 | 0.12 |
| Collateral damage fraction (DBH ≥ $DBH_{min}$) | 0.036 | 0.018 | 0.024 | 0.012 |
| mechanical damage fraction (DBH < $DBH_{max\_infra}$[2]) | 0.113 | 0.073 | 0.033 | 0.024 |
| Understory death fraction[3] | 0.65 | 0.65 | 0.65 | 0.65 |

[1]$DBH_{min}$ = 50 cm
[2]$DBH_{max\_infra}$ = 30 cm
[3]Applied to the new patch generated by direct felling and collateral damage





Table 4. Comparison of energy fluxes (Mean ± Standard Deviation) between eddy covariance
tower measurements and FATES simulations.

| Variables | LH (W m$^{-2}$) | SH (W m$^{-2}$) | Rn (W m$^{-2}$) |
|---|---|---|---|
| **Observed (km83)** | **101.6±8.0** | **25.6±5.2** | **129.3±18.5** |
| Simulated (Intact) | 108.6±21.0 | 20.1±24.7 | 128.8±15.6 |
| **Simulated (RIL$_{low}$)** | **108.3±20.8** | **20.5±24.3** | **128.9±15.5** |
| Simulated (RIL$_{high}$) | 108.0±20.5 | 20.9±23.9 | 129.0±15.4 |
| Simulated (CL$_{low}$) | 108.3±20.8 | 20.5±24.3 | 128.9±15.5 |
| Simulated (CL$_{high}$) | 108.0±20.5 | 20.5±24.3 | 129.0±15.4 |





Table 5. Comparison of carbon budget terms between observation-based estimates[*] and simulations at km83

| Variable | Obs. | | Simulated | | | | | | | | |
|---|---|---|---|---|---|---|---|---|---|---|---|
| | Pre-logging | 3-yr Post-logging | Intact | Disturb level | 0 yr | 1 yr | 3 yr | 15 yr | 30 yr | 50 yr | 70 yr |
| AGB (MgC ha$^{-1}$) | 165 | 147 | 155 | RIL$_{low}$ | 138 | 138 | 140 | 143 | 146 | 150 | 155 |
| | | | | RIL$_{high}$ | 119 | 120 | 122 | 127 | 133 | 141 | 147 |
| | | | | CL$_{low}$ | 138 | 138 | 139 | 143 | 145 | 149 | 153 |
| | | | | CL$_{high}$ | 119 | 119 | 121 | 128 | 133 | 140 | 146 |
| Necromass (MgC ha$^{-1}$) | 58.4 | 74.4 | 41.1 | RIL$_{low}$ | 59.6 | 55.3 | 48.5 | 41.2 | 41.0 | 41.4 | 41.4 |
| | | | | RIL$_{high}$ | 79.5 | 70.3 | 56.9 | 41.2 | 39.8 | 39.4 | 41.6 |
| | | | | CL$_{low}$ | 60.0 | 55.4 | 48.7 | 41.0 | 40.9 | 40.9 | 41.1 |
| | | | | CL$_{high}$ | 80.1 | 70.6 | 57.2 | 40.3 | 39.3 | 40.7 | 41.0 |
| NEE (MgC ha$^{-1}$ yr$^{-1}$) | -0.6±0.8 | -1.0±0.7 | 0.49 | RIL$_{low}$ | 0.54 | 1.76 | 1.53 | 0.57 | 0.23 | 0.22 | 0.33 |
| | | | | RIL$_{high}$ | 0.59 | 3.94 | 2.62 | 0.64 | 0.15 | 0.08 | 0.26 |
| | | | | CL$_{low}$ | 0.54 | 1.83 | 1.55 | 0.56 | 0.24 | 0.31 | 0.44 |
| | | | | CL$_{high}$ | 0.59 | 4.09 | 2.66 | 0.59 | 0.07 | 0.16 | 0.28 |
| GPP (MgC ha$^{-1}$ yr$^{-1}$) | 32.6±1.3 | 32.0±1.3 | 17.6 | RIL$_{low}$ | 17.5 | 16.7 | 17.9 | 17.3 | 17.7 | 17.2 | 17.2 |
| | | | | RIL$_{high}$ | 17.3 | 15.7 | 17.9 | 17.5 | 17.9 | 17.3 | 17.2 |
| | | | | CL$_{low}$ | 17.5 | 16.6 | 17.9 | 17.3 | 17.7 | 17.3 | 17.2 |
| | | | | CL$_{high}$ | 17.3 | 15.4 | 17.8 | 17.6 | 18.0 | 17.3 | 17.2 |
| NPP (MgC ha$^{-1}$ yr$^{-1}$) | 9.5 | 9.8 | 8.9 | RIL$_{low}$ | 8.9 | 8.6 | 9.2 | 8.6 | 9.0 | 8.7 | 8.6 |
| | | | | RIL$_{high}$ | 8.9 | 8.1 | 9.3 | 8.8 | 9.1 | 8.7 | 8.7 |
| | | | | CL$_{low}$ | 8.9 | 8.5 | 9.2 | 8.7 | 9.0 | 8.6 | 8.6 |
| | | | | CL$_{high}$ | 8.9 | 7.9 | 9.3 | 8.7 | 9.1 | 8.7 | 8.7 |
| ER (MgC ha$^{-1}$ yr$^{-1}$) | 31.9±1.7 | 31.0±1.6 | 18.1 | RIL$_{low}$ | 18.0 | 18.6 | 19.6 | 17.9 | 18.0 | 17.5 | 17.5 |
| | | | | RIL$_{high}$ | 17.9 | 19.6 | 21.2 | 18.2 | 18.1 | 17.3 | 17.5 |
| | | | | CL$_{low}$ | 18.0 | 18.6 | 19.6 | 17.9 | 18.0 | 17.5 | 17.7 |
| | | | | CL$_{high}$ | 17.9 | 19.5 | 21.2 | 18.2 | 18.0 | 17.4 | 17.5 |
| HR (MgC ha$^{-1}$ yr$^{-1}$) | 8.9 | 10.4 | 9.4 | RIL$_{low}$ | 9.4 | 10.5 | 10.9 | 9.2 | 9.3 | 8.9 | 8.9 |
| | | | | RIL$_{high}$ | 9.5 | 12.0 | 12.6 | 9.4 | 9.3 | 8.8 | 8.9 |
| | | | | CL$_{low}$ | 9.4 | 10.5 | 11.0 | 9.2 | 9.3 | 8.9 | 9.0 |
| | | | | CL$_{high}$ | 9.5 | 12.0 | 12.6 | 9.3 | 9.2 | 8.9 | 9.0 |
| AR (MgC ha$^{-1}$ yr$^{-1}$) | 23.1 | 20.1 | 8.7 | RIL$_{low}$ | 8.6 | 8.1 | 8.7 | 8.7 | 8.7 | 8.5 | 8.6 |
| | | | | RIL$_{high}$ | 8.4 | 7.6 | 8.6 | 8.8 | 8.8 | 8.6 | 8.6 |
| | | | | CL$_{low}$ | 8.5 | 8.1 | 8.7 | 8.7 | 8.7 | 8.5 | 8.7 |
| | | | | CL$_{high}$ | 8.4 | 7.5 | 8.6 | 8.9 | 8.8 | 8.5 | 8.6 |

[*]Source of observation-based estimates: Miller et al. (2011), Uncertainty in carbon fluxes (GPP, ER, NEE) are based on u*-filter cutoff analyses described in the same paper.





Table 6. Simulated Stem Density (N ha$^{-1}$) Distribution at km83.

| Years following logging | Disturbance level | Size classes (DBH, cm) | | | |
|---|---|---|---|---|---|
| | | < 10 cm | 10-30 cm | 30-50 cm | ≥ 50 cm |
| Pre-logging | Intact | 9737 | 145 | 43 | 44 |
| 0-yr | $RIL_{low}$ | 9203 | 143 | 43 | 38 |
| | $RIL_{high}$ | 8813 | 142 | 43 | 32 |
| | $CL_{low}$ | 8694 | 138 | 43 | 38 |
| | $CL_{high}$ | 8012 | 134 | 43 | 32 |
| 1-yr | $RIL_{low}$ | 9072 | 147 | 28 | 51 |
| | $RIL_{high}$ | 9113 | 146 | 28 | 45 |
| | $CL_{low}$ | 9159 | 143 | 28 | 51 |
| | $CL_{high}$ | 9083 | 139 | 28 | 45 |
| 3-yr | $RIL_{low}$ | 9909 | 139 | 25 | 47 |
| | $RIL_{high}$ | 11010 | 136 | 27 | 41 |
| | $CL_{low}$ | 9676 | 121 | 25 | 47 |
| | $CL_{high}$ | 10659 | 115 | 28 | 41 |
| 15-yr | $RIL_{low}$ | 8609 | 161 | 66 | 37 |
| | $RIL_{high}$ | 8526 | 188 | 66 | 33 |
| | $CL_{low}$ | 6698 | 171 | 64 | 37 |
| | $CL_{high}$ | 8787 | 248 | 62 | 33 |
| 30-yr | $RIL_{low}$ | 9277 | 90 | 68 | 33 |
| | $RIL_{high}$ | 9225 | 118 | 90 | 31 |
| | $CL_{low}$ | 8132 | 140 | 69 | 33 |
| | $CL_{high}$ | 10128 | 101 | 85 | 31 |
| 50-yr | $RIL_{low}$ | 6995 | 132 | 64 | 54 |
| | $RIL_{high}$ | 8196 | 129 | 16 | 62 |
| | $CL_{low}$ | 8336 | 125 | 21 | 55 |
| | $CL_{high}$ | 7487 | 110 | 59 | 59 |
| 70-yr | $RIL_{low}$ | 7904 | 128 | 55 | 52 |
| | $RIL_{high}$ | 6248 | 83 | 11 | 67 |
| | $CL_{low}$ | 9352 | 149 | 31 | 58 |
| | $CL_{high}$ | 6589 | 202 | 31 | 55 |





Table 7. Simulated Basal Area (m² ha⁻¹) Distribution at km83.

| Years following logging | Disturbance level | Size classes (DBH, cm) | | | |
|---|---|---|---|---|---|
| | | < 10 cm | 10-30 cm | 30-50 cm | ≥ 50 cm |
| Pre-logging | Intact | 0.9 | 3.4 | 5.3 | 41.6 |
| 0-yr | $RIL_{low}$ | 0.9 | 3.3 | 5.2 | 36.3 |
| | $RIL_{high}$ | 0.8 | 3.3 | 5.2 | 30.5 |
| | $CL_{low}$ | 0.8 | 3.2 | 5.3 | 36.3 |
| | $CL_{high}$ | 0.8 | 3.2 | 5.3 | 30.5 |
| 1-yr | $RIL_{low}$ | 0.7 | 3.7 | 2.5 | 38.9 |
| | $RIL_{high}$ | 0.7 | 3.7 | 2.5 | 33.1 |
| | $CL_{low}$ | 0.8 | 3.6 | 2.5 | 38.9 |
| | $CL_{high}$ | 0.8 | 3.5 | 2.5 | 33.1 |
| 3-yr | $RIL_{low}$ | 1.1 | 4.3 | 2.6 | 38.4 |
| | $RIL_{high}$ | 1.4 | 4.2 | 2.7 | 32.7 |
| | $CL_{low}$ | 1.3 | 4.0 | 2.6 | 38.4 |
| | $CL_{high}$ | 1.6 | 3.7 | 2.8 | 32.7 |
| 15-yr | $RIL_{low}$ | 0.7 | 3.6 | 7.0 | 36.3 |
| | $RIL_{high}$ | 1.1 | 4.1 | 6.9 | 31.2 |
| | $CL_{low}$ | 0.6 | 3.8 | 6.8 | 36.4 |
| | $CL_{high}$ | 0.6 | 5.0 | 6.6 | 31.2 |
| 30-yr | $RIL_{low}$ | 1.1 | 3.1 | 8.3 | 35.6 |
| | $RIL_{high}$ | 1.0 | 2.6 | 10.3 | 31.2 |
| | $CL_{low}$ | 0.8 | 3.5 | 8.3 | 35.6 |
| | $CL_{high}$ | 1.1 | 3.1 | 9.7 | 31.2 |
| 50-yr | $RIL_{low}$ | 0.4 | 2.2 | 6.1 | 40.5 |
| | $RIL_{high}$ | 0.7 | 5.0 | 2.2 | 39.3 |
| | $CL_{low}$ | 1.0 | 4.6 | 2.9 | 40.7 |
| | $CL_{high}$ | 0.6 | 2.3 | 5.5 | 38.5 |
| 70-yr | $RIL_{low}$ | 0.5 | 2.2 | 6.5 | 41.6 |
| | $RIL_{high}$ | 1.2 | 4.0 | 1.0 | 42.4 |
| | $CL_{low}$ | 0.7 | 3.6 | 3.0 | 42.9 |
| | $CL_{high}$ | 0.4 | 4.1 | 3.8 | 40.1 |





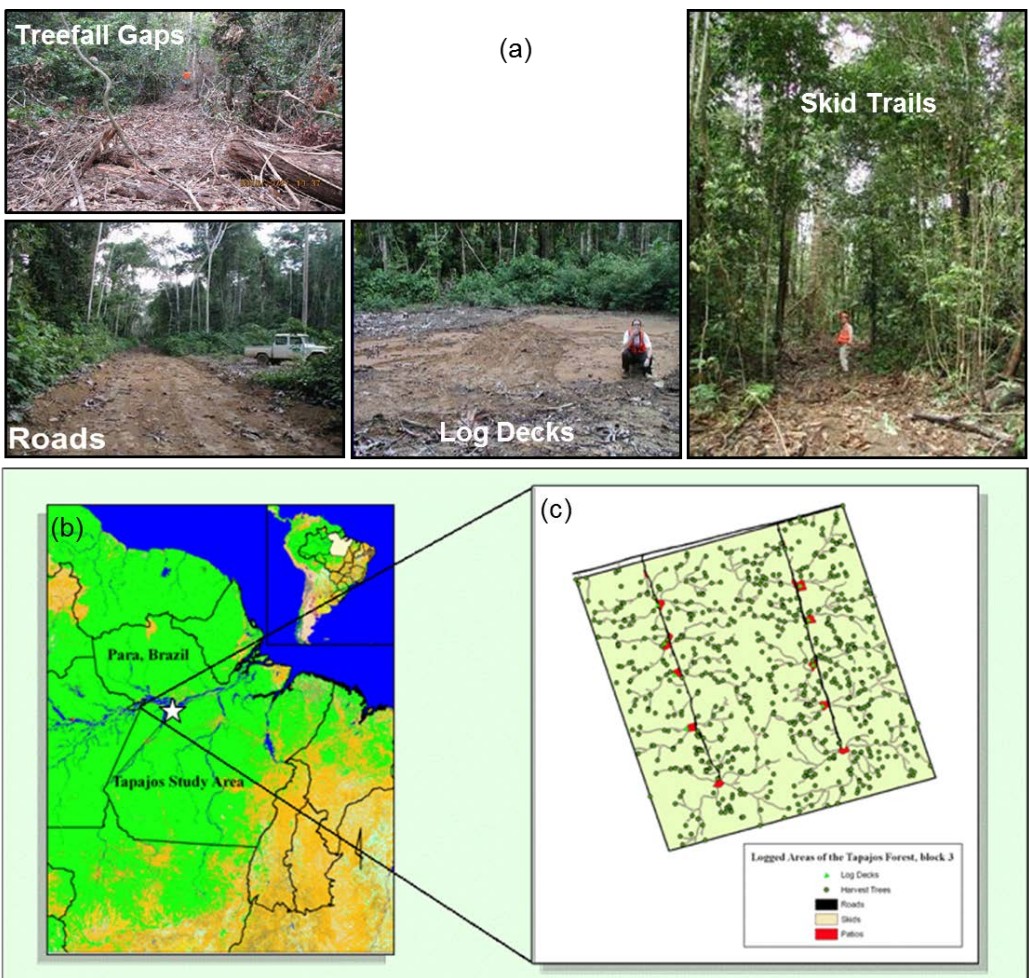

Figure 1. (a) Landscape components of selective logging; (b) location of the Tapajos National Forest in the
Amazon; and (c) a typical logging block showing tree-fall location, skid trail, road, and log deck coverages.
Panels (b) and (c) are from *Asner et al*. (2008).



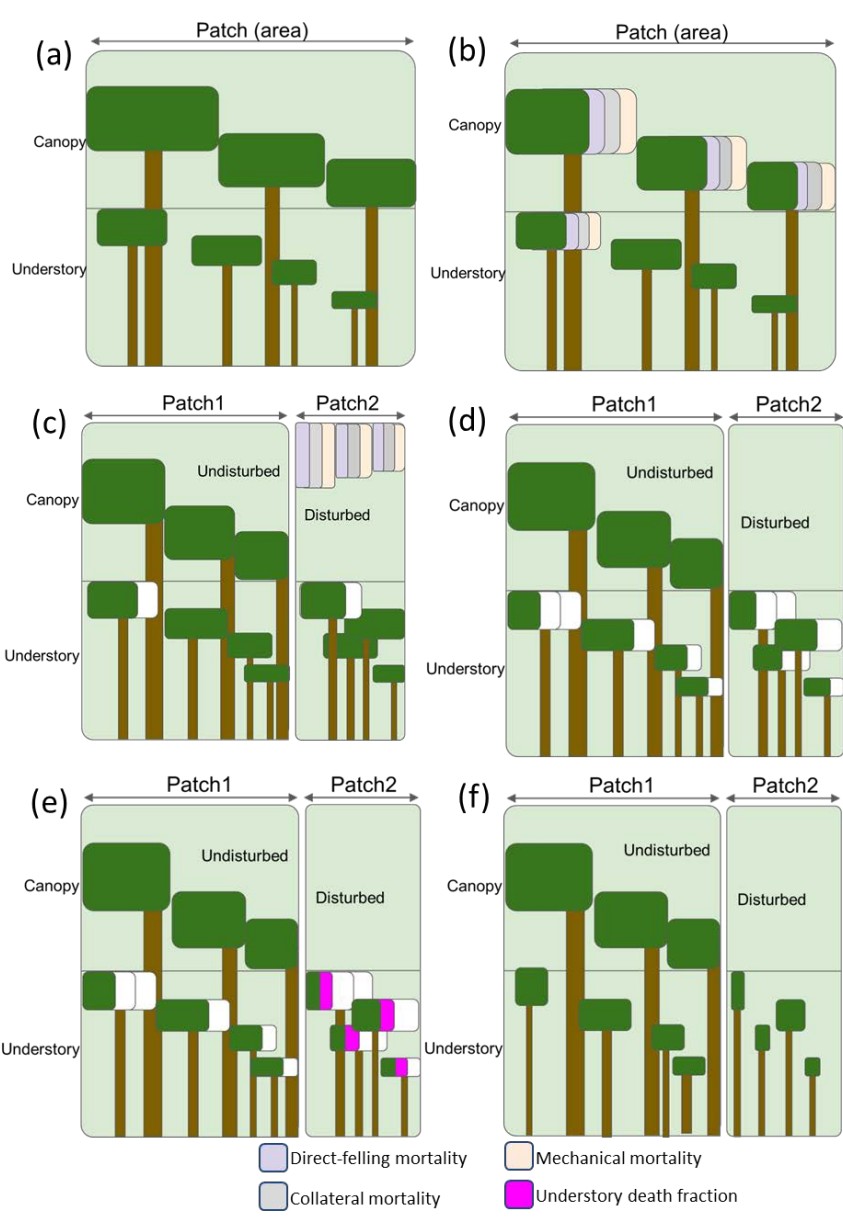

Figure 2. The mortality types (direct-felling, mechanical, and collateral) and patch generating process in the FATES logging module. The white fraction in (c), (d), (f) indicates mortality associated with other disturbances in FATES. (a) Canopy and understory layers in each cohort in FATES; (b) Mortality applied at the time of a logging event; (c) the patch fission process following a given logging event; (d) canopy removal in the disturbed patch following the logging event; (e) calculate the understory survivorship based on the understory death fraction in each patch; (d) the final states of the intact and disturbed patches.



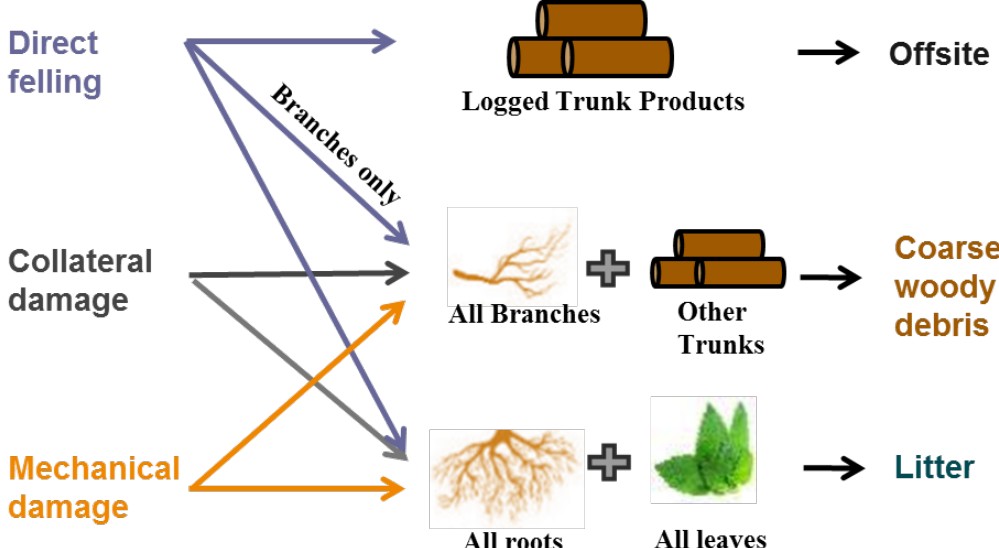

43    Figure 3. The flow of necromass following logging.




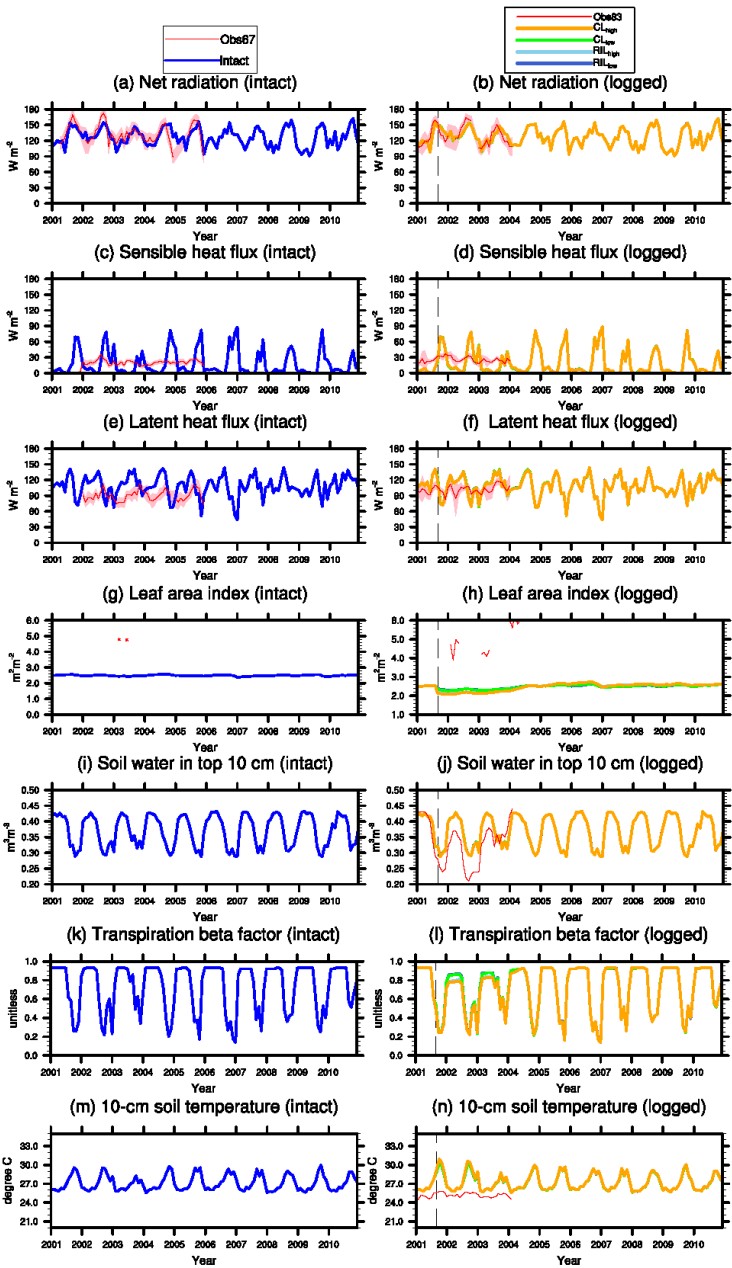

Figure 4. Simulated energy budget terms and leaf area indices in intact and logged forests compared to observations from km67 (left) and km83 (right) (*Miller et al.*, 2011). The dashed vertical line indicates the timing of the logging event. The shaded area in panel (a)-(f) are uncertainty estimates based on based on u*-filter cutoff analyses in *Miller et al.* (2011).

50




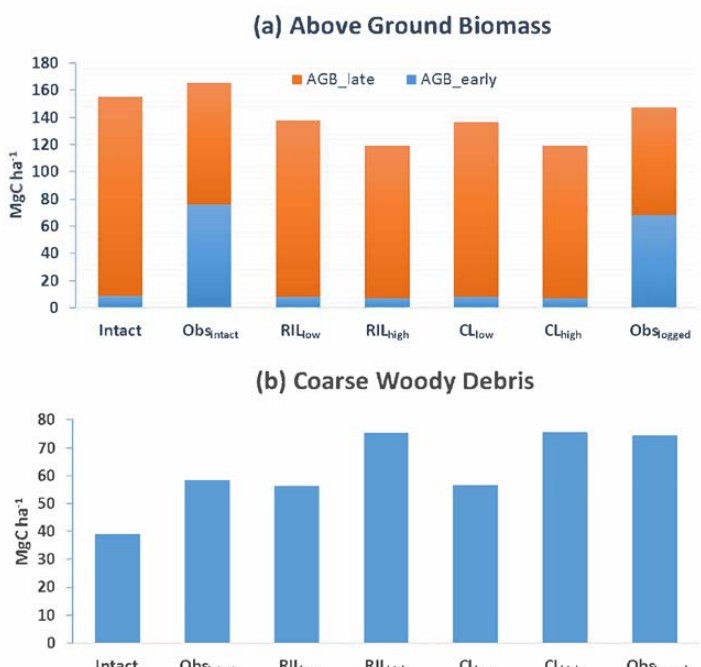

Figure 5. Simulated (a) Above Ground Biomass; and (b) Coarse Woody Debris in intact and logged forests in a one-year period before or after the logging event in the four logging scenarios listed in Table 3. The observations (Obs$_{intact}$ and Obs$_{logged}$) were derived from inventory (*Menton et al.*, 2011; *de Sousa et al.*, 2011).



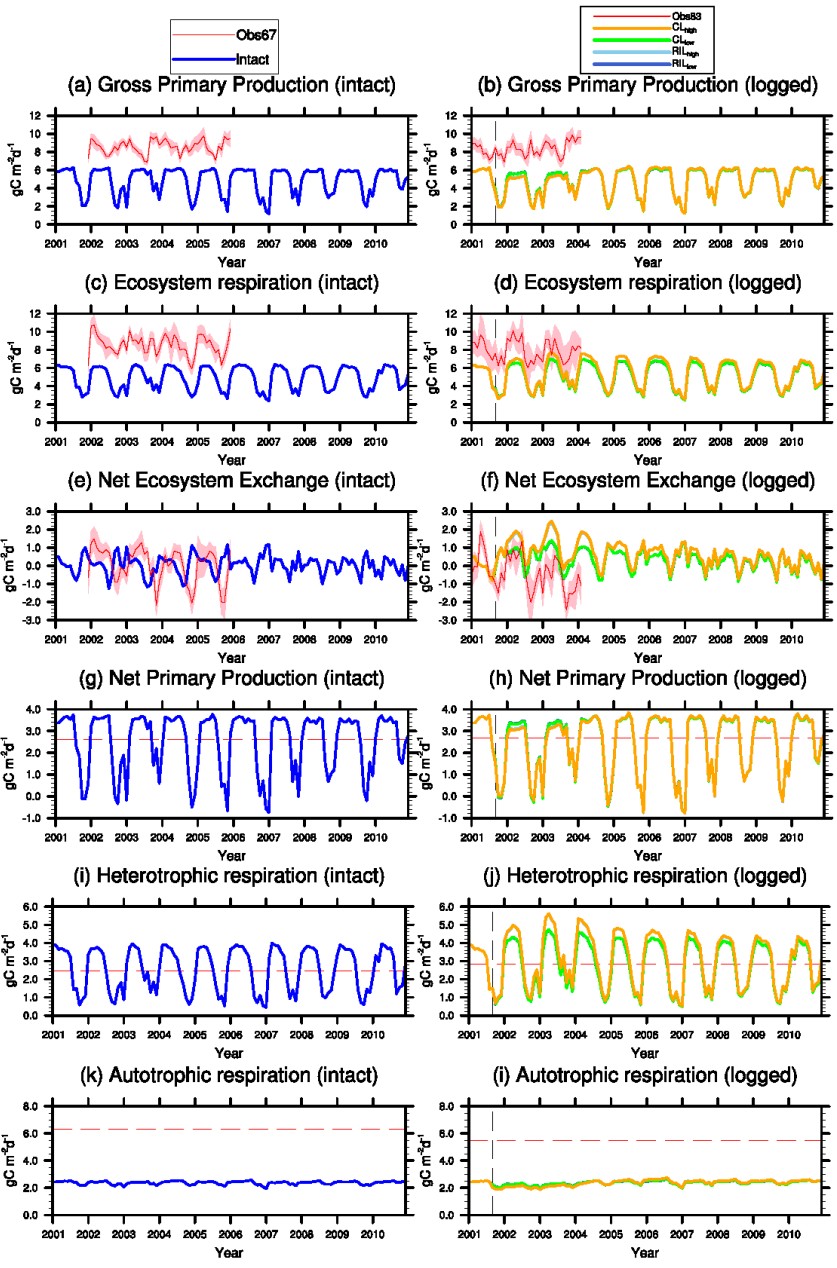


Figure 6. Simulated carbon fluxes in intact and logged forests compared to observed fluxes from km67
(left) and km83 (right). The dashed black vertical line indicates the timing of the logging event, while the
red dashed horizontal line indicates estimated fluxes derived based on eddy covariance measurements and
inventory (*Miller et al.*, 2011). The shaded area in panel (a)-(f) are uncertainty estimates based on based on
u*-filter cutoff analyses in *Miller et al.* (2011).




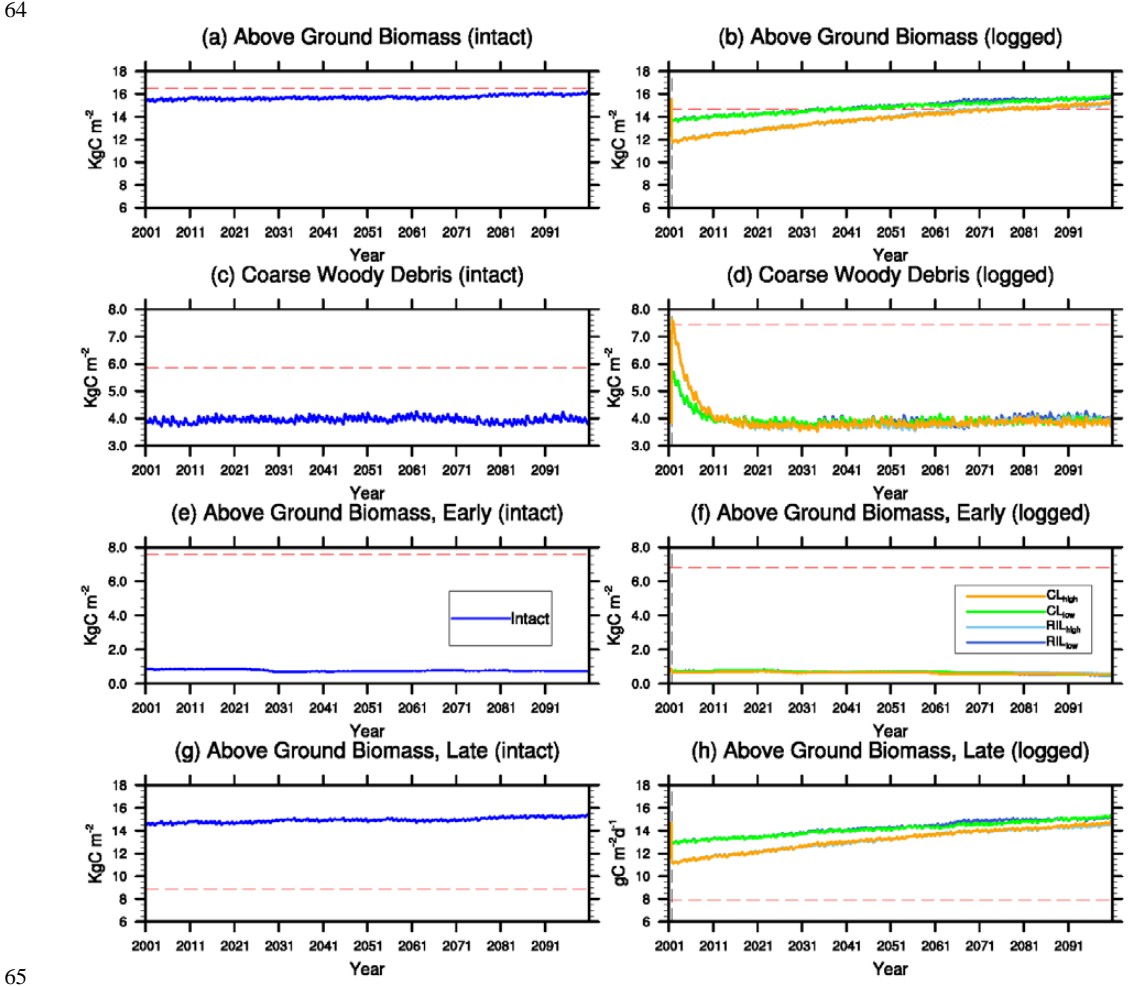


Figure 7. Trajectories of carbon pools in intact (left) and logged (right) forests. The dashed black vertical
line indicates the timing of the logging event. The red dashed horizontal line indicates observed pre- (left)
and post-logging (right) inventories respectively (*Menton et al.*, 2011; de *Sousa et al.*, 2011).




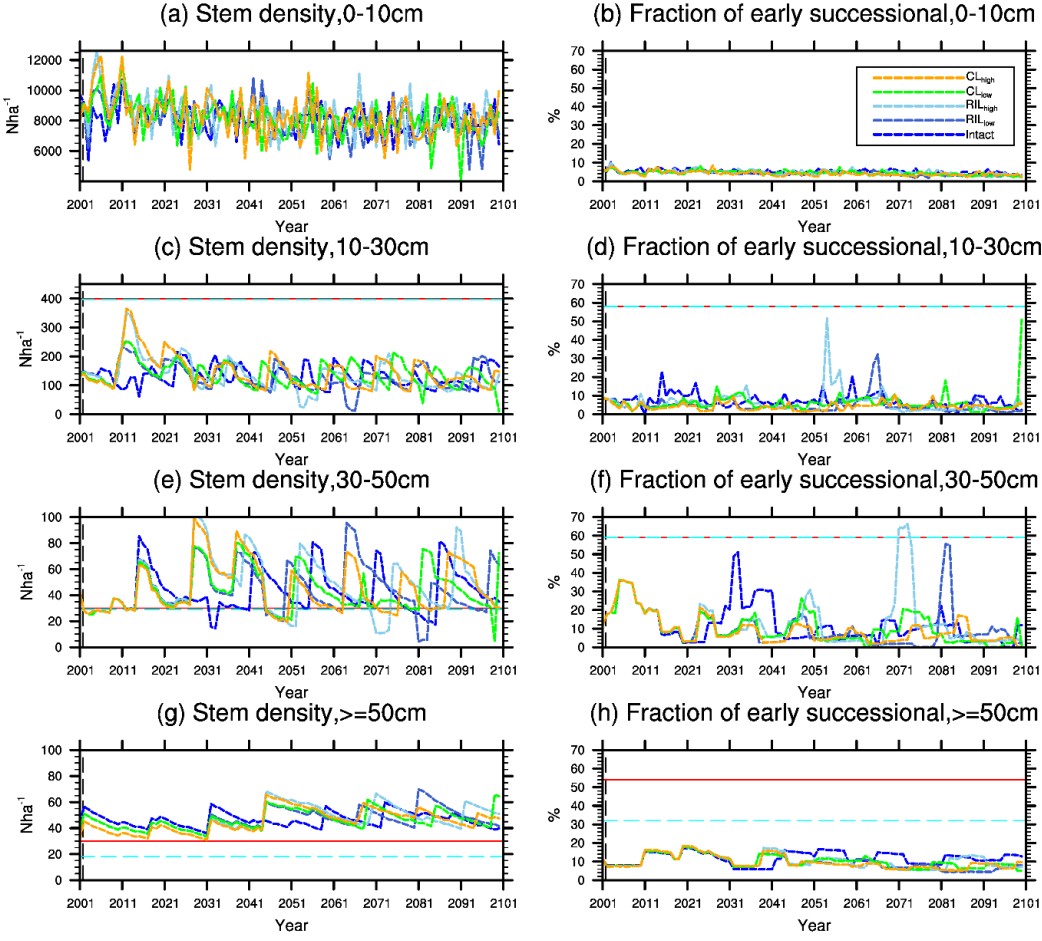


Figure 8. Changes in total stem densities and the fractions of the early successional PFT in different size classes following a single logging event on 1 September 2001 at km83. The black dashed vertical line indicates the timing of the logging event, while the red solid line and the cyan dashed horizontal line indicate observed pre- and post-logging inventories respectively (*Menton et al.*, 2011; *de Sousa et al.*, 2011).







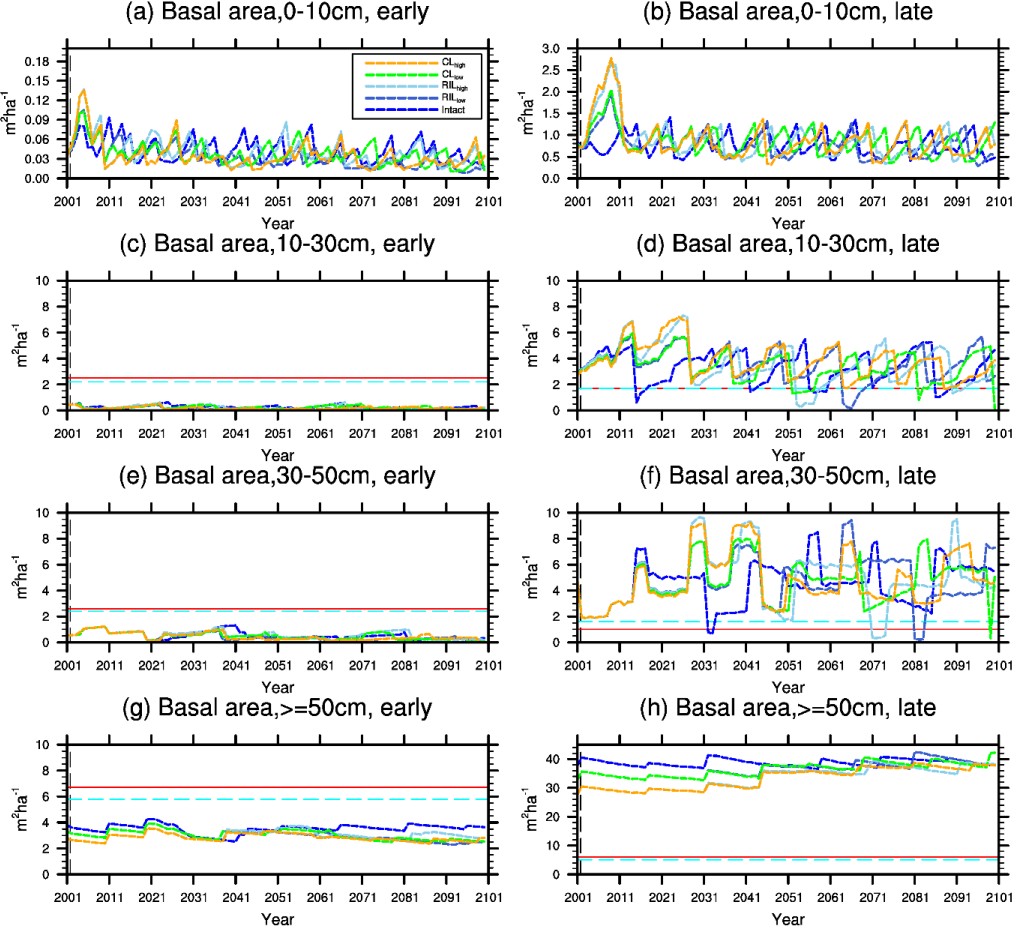

Figure 9. Changes in basal area of the two PFTs in different size classes following a single logging event on 1 September 2001 at km83. The black dashed vertical line indicates the timing of the logging event, while the red solid line and the cyan dashed horizontal line indicates observed pre- and post-logging inventories respectively (*Menton et al.*, 2011; *de Sousa et al.*, 2011). Note that for the size class 0-10 cm, observations are not available from the inventory.





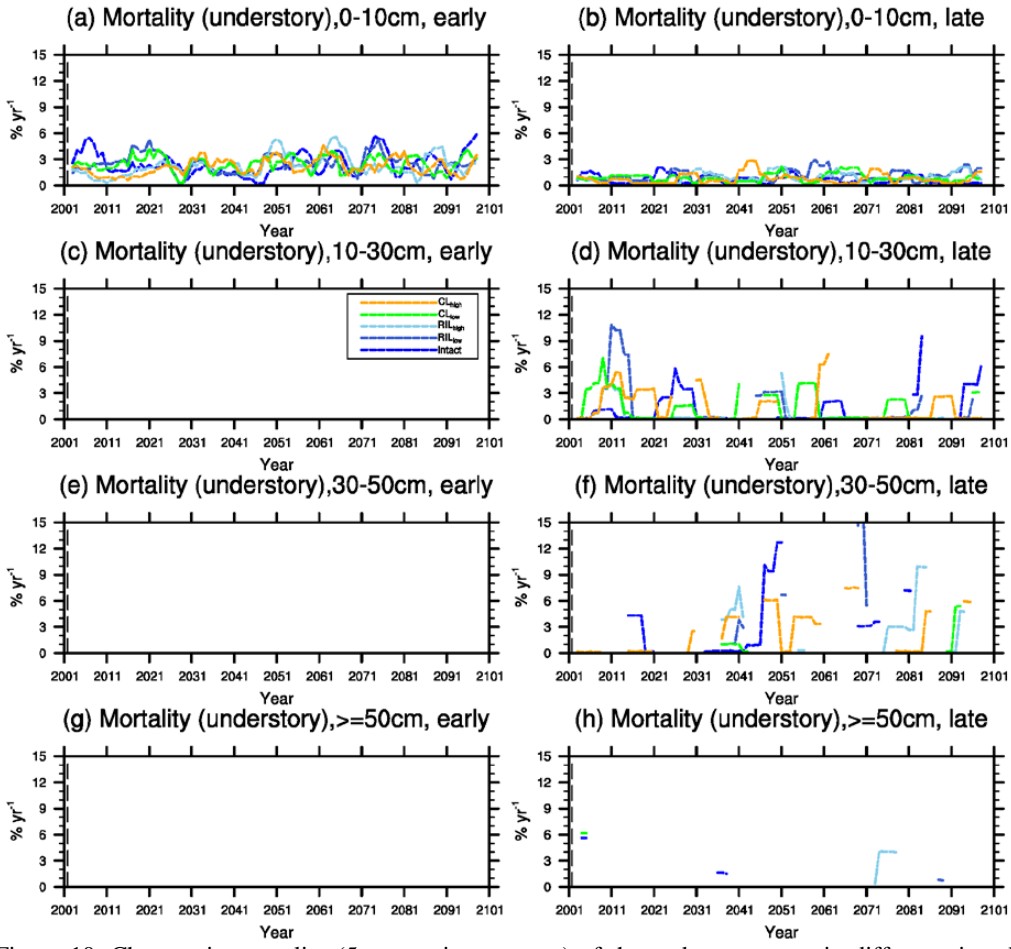

Figure 10. Changes in mortality (5-yr running average) of the understory trees in different size classes
following a single logging event on 1 September 2001. The black dashed vertical line indicates the timing
of the logging event. Note that mortality is not defined in large size classes because no tree survives in the
understory or effectively promoted as canopy trees, especially for the early successional PFT.





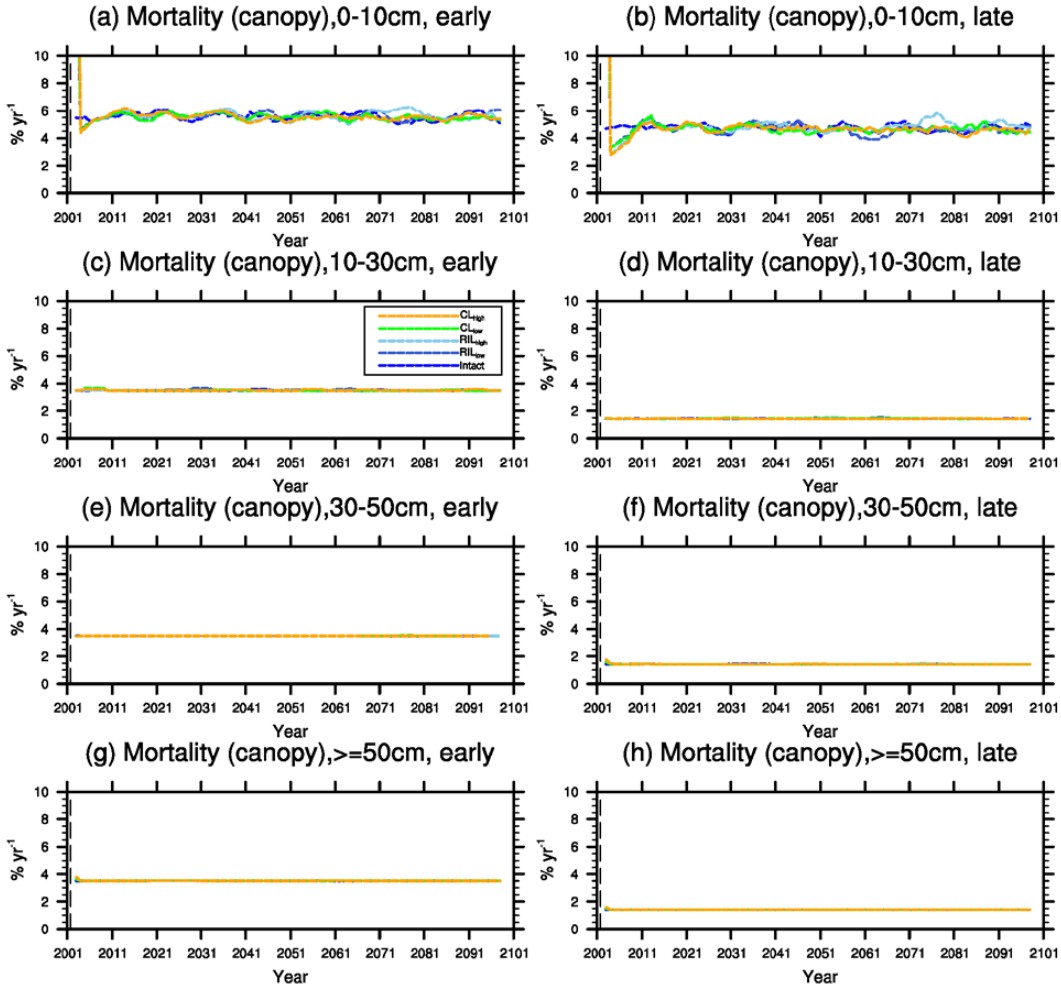

Figure 11. Changes in mortality (5-yr running average) of the canopy trees in different size classes
following a single logging event on 1 September 2001. The black dashed vertical line indicates the timing
of the logging event.