# Peer review of "Assessing impacts of selective logging on water,"

_Biogeosciences, 2019_

## Referee Comment (RC1) · Anonymous Referee #1 · 20 May 2019

General Comments: This is an excellently written manuscript, very readable, and all arguments and assumptions are clearly stated. The work is timely as there is a general deficiency amongst models (especially biogeochemical) to have the ability to reflect managed disturbances, especially partial disturbance such as thinning or selective harvest. This will also be useful for disturbance through beetle kill and drought as there are many post-disturbance structural and successional changes/trajectories that need better representation in models. The correct representation of the immediate pool

changes (CWD, litter, etc) are essential.

Specific comments: 1. The other pool that is often neglected is the dead tree pool (snags; standing dead wood). I understand that addition of this pool would require a revision to FATES (not trivial) but harvest operations (especially thinning) can lead to live tree death from machine damage and windthrow. This will be more important for using FATES in temperate, coniferous systems and the varied biogeochemical legacy of standing versus downed wood is important (Edburg et al. 2011, Edburg et al. 2012). Maybe this could be mentioned in the discussion for future model development? 2. The results for GPP and NPP recovery are interesting. It is my understanding though that there is no Nitrogen limitation on growth in FATES (versus CLM; the non-ED version). The model is underestimating GPP and AR and in this case, it is not because of N limitation (in the model). It appears it is low LAI; if this is 'fixed' do you think GPP may then be overestimated and there will be issues with non-modeled nutrient limitation? Just something to think about.

Edburg, S. L., J. A. Hicke, P. D. Brooks, E. G. Pendall, B. E. Ewers, U. Norton, D. Gochis, E. D. Gutmann, and A. J. H. Meddens. 2012. Cascading impacts of bark beetle-caused tree mortality on coupled biogeophysical and biogeochemical processes. Frontiers in Ecology and the Environment 10:416-424. Edburg, S. L., J. A. Hicke, D. M. Lawrence, and P. E. Thornton. 2011. Simulating coupled carbon and nitrogen dynamics following mountain pine beetle outbreaks in the western United States. Journal of Geophysical Research-Biogeosciences 116.

---

## Referee Comment (RC2) · Anonymous Referee #2 · 23 May 2019

The authors parameterized two PFTs in FATES for a tropical forest site and embedded a selective logging module. As a model description paper, the manuscript appears fairly complete and informative for others interested in understanding the model design better. The authors present results of a calibration exercise at the two sites by comparing simulated and observed responses to logging at one site, and comparing it to undisturbed dynamics at the second site. The results show that the model is modestly successful in capturing some facets of the forest/ecosystem dynamics, but performs

poorly at others. As a biogeoscience paper, we think the manuscript falls disappointingly short of reaching some interesting potential for insight.

Specifically, there is a substantial mismatch between data and the model for some very basic forest/ecosystem characteristics. There are large errors in LAI, GPP, RH, and age structure, as conceded by the authors. Even for the control site, GPP shows an almost opposite seasonality between model and data. The errors caused by calibration are much greater than the variation due to disturbance levels (Table 5). While it would have been preferable to have a more successful calibration, falling short of that, the authors should present a coherent and robust explanation of what the fundamental structural problems were, with figures specifically illustrating the insights. That would elevate the significance of the paper, and increase its utility for those seeking to do similar work.

Comments for each section Introduction: The authors lay out pertinent background information but the text does not explicitly articulate a cogent and compelling argument for why this study is needed. I think the introduction would be more effective if text were added to make the connection between the background information and the aims of the paper.

Methods: They report that FATES is very sensitive to parameter values, to a point that with some combinations the two PFTs cannot coexist. That is somewhat worrisome. There is a fair amount of detail given on how a logging activity is applied to a patch, but it's a bit unclear which patches are selected for logging.

Results: Given that SH mismatch happens at the seasonal scale it would be useful to have some analysis results at that time scale. For example, the results in Fig 4 could be replotted at the seasonal scale (average across years). Low SH is attributed to attributed to low LAI, but what's causing LAI? It seems a fairly straight forward question to answer (or at least speculate). It appears they did not go far enough with the most interesting/instructive part of the exploration. Similarly with soil moisture, the authors present a cursory analysis of soil water uptake. What about SWC of the deeper

layer(s)? What is their relation to simulated ET?

Line-by-line comments

56: "suggested a net tropical forest land-use source of 1.3 . . ." is grammatically incorrect. I suggest something like, "suggested tropical forests can be a net source of 1.3 . . . from land-use change."

63: The authors defined degradation as widespread damage to remaining trees, subcanopy vegetation and soils, and that it could cause as much as 40% carbon loss of clearcut deforestation. In your simulation, how did you define the effect of degradation?

66: delete "as".

67: hyphenate one-eighth.

70: Extraneous parenthesis.

78: couple terrestrial and atmospheric. . .

78-80: Perhaps list some examples of those models?

79: comma after "change"

81: representation of wood harvest. . .

83: Is that in LM3V? It's unclear.

86-89: It would be better to define selective logging earlier, since it is referred to many times prior to this point in the text.

90: Not just simplified but absent

91: did not

98: "tremendous" is overly dramatic

108: "assess the simulated recovery of Tapajos National Forest. . ."

[Figure]

109: summary of

113: simulated forest trajectory

120: The authors describe FATES model as a further developed model based on CLM (ED), which can be viewed as an early version of FATES. Then, which version is what you used in this study? Is there any paper that formerly published FATES model?

152: Specific should be lowercase. But lines 150-155 is a run-on sentence. They need to at least insert a conjunction.

162: Delete hyphen in co-existence

164: Delete hyphen in co-existence

176-7: "transports off-site by adding…" should be "transports off-site by reducing site carbon pools. Remaining necromass … are added to coarse woody debris and litter pools."

181: "are represented" should be "are conceptually represented." Because this paragraph just talks about the various concepts, and not specific implementation of logging regimes & effects.

189: It's unclear if FATES implements these two types of logging practices.

221-4: Parentheses seem unnecessary.

227: "…whose…" is grammatically incorrect.

250: Delete respectively.

273: Awkward phrasing. I recommend, "To … conservation, we calculate,". And then say how del-B is used to ensure mass conservation. (Just calculating del-B doesn't ensure mass conservation).

315: Equation should be plural. Or, if singular, use an article.

327: . . . this forest comprised 399 . . .

328: Replace semicolon with a period.

339: This is the first use of the name "CLM(FATES)". I think authors should explain this means running FATES as a module inside CLM, then use the name "CLM(FATES)".

341: "km 67" -> "km67"

342: "km 83" -> "km83"

344: "be covered by bare soil" -> "be bare soil."

346: "even with the" -> "even within the"

362: "specified" -> applied, prescribed or implemented

363: "following" doesn't work here, since authors describe an RIL in the sentence.

447: "Table 5" should be "Table 6."

455: Could it also be related to how crowns are represented (leaf area exists at a single height), and how light is allowed to penetrate downward?

Tbl 2: In caption, m2 should have 2 as a superscript. Also, it would make more sense to have subtotals below numbers for DBH-based classes. There is an asterisk and a footnote but I don't see the corresponding asterisk in the caption text. I think the whole footnote could just go in the caption.

Tbl 5: This is a difficult way to compare values. It would be much better to present as a graph. The caption should spell out all the abbreviations used.

Fig 1: Legend of panel C is too small and fuzzy and therefore nearly illegible.

Fig 2: In panel (c), having mortality symbols line up at the top was a little confusing to me, visually. Recommend positioning them at the canopy level. The caption for (e) should read, "calculating. . ." The caption for (f) is mislabeled as "(d)".

Fig 4: In the PDF version I got, the font used in the figure is too small, and appears with jagged edges, even when magnified. The phrase "based on" is repeated, and should be deleted. The legend for the panels on right are confusing. I think the caption should clearly say four types of logging effects are plotted. The caption should also state that there are no observation values for panels I, k, l, and m.

Fig 5: Above ground biomass and coarse woody debris should not be capitalized. I recommend coloring the observation bars differently, to make them stand out. AGB_early and AGB_late should be explained in the caption.

Fig 6: In the caption, "derived based on" should be "derived from". Do the red dashed horizontal lines represent period averages?

Fig 8, 9: In most of the panels, the intact curve is hard to see, and hard to distinguish from RIL-low. I suggest using a solid black line, and draw it on top of other lines.

---

## Author Response (AR1)

**Assessing impacts of selective logging on water, energy, and carbon budgets and ecosystem dynamics in Amazon forests using the Functionally Assembled Terrestrial Ecosystem Simulator [MS No.: bg-2019-129]**

**Responses to review comments**

**Anonymous Referee #1:**

***General Comments:*** *This is an excellently written manuscript, very readable, and all arguments and assumptions are clearly stated. The work is timely as there is a general deficiency amongst models (especially biogeochemical) to have the ability to reflect managed disturbances, especially partial disturbance such as thinning or selective harvest. This will also be useful for disturbance through beetle kill and drought as there are many post-disturbance structural and successional changes/trajectories that need better representation in models. The correct representation of the immediate pool.*

**Response**:

Thank you very much for the positive comments on our work and we are extremely encouraged to continue developing the model to represent other key ecosystem disturbances that are enabled by this new development in FATES

***Specific comments:***

*1. The other pool that is often neglected is the dead tree pool (snags; standing dead wood). I understand that addition of this pool would require a revision to FATES (not trivial) but harvest operations (especially thinning) can lead to live tree death from machine damage and windthrow. This will be more important for using FATES in temperate, coniferous systems and the varied biogeochemical legacy of standing versus downed wood is important (Edburg et al. 2011, Edburg et al. 2012). Maybe this could be mentioned in the discussion for future model development?*

**Response**:

We will include discussions on the potential and challenges to incorporate a dead tree pool to facilitate the application of FATES to in other ecosystems in the revised manuscript.

*2. The results for GPP and NPP recovery are interesting. It is my understanding though that there is no Nitrogen limitation on growth in FATES (versus CLM; the non-ED version). The model is underestimating GPP and AR and in this case, it is not because of N limitation (in the model). It appears it is low LAI; if this is 'fixed' do you think GPP may*

*then be overestimated and there will be issues with non-modeled nutrient limitation? Just something to think about.*

Edburg, S. L., J. A. Hicke, P. D. Brooks, E. G. Pendall, B. E. Ewers, U. Norton, D. Gochis, E. D. Gutmann, and A. J. H. Meddens. 2012. Cascading impacts of bark beetle-caused tree mortality on coupled biogeophysical and biogeochemical processes. Frontiers in Ecology and the Environment 10:416-424. Edburg, S. L., J. A. Hicke, D. M. Lawrence, and P. E. Thornton. 2011. Simulating coupled carbon and nitrogen dynamics following mountain pine beetle outbreaks in the western United States. Journal of Geophysical Research-Biogeosciences 116.

**Response**:

In the revision, we have updated the model to a new version of FATES in which the penalty for establishing leaf biomass is greatly reduced. We also performed ensemble simulations to evaluate potential ways to improve the low LAI bias by perturbing key physiological parameters. We have revised the manuscript to incorporate the new results and more discussions along this line.

**Assessing impacts of selective logging on water, energy, and carbon budgets and ecosystem dynamics in Amazon forests using the Functionally Assembled Terrestrial Ecosystem Simulator [MS No.: bg-2019-129]**

**Responses to review comments**

**Anonymous Referee #2:**

*The authors parameterized two PFTs in FATES for a tropical forest site and embedded a selective logging module. As a model description paper, the manuscript appears fairly complete and informative for others interested in understanding the model design better. The authors present results of a calibration exercise at the two sites by comparing simulated and observed responses to logging at one site, and comparing it to undisturbed dynamics at the second site. The results show that the model is modestly successful in capturing some facets of the forest/ecosystem dynamics, but performs poorly at others.*

**Response**:

Thanks for the nice summary. We agree with the referee that the model can be improved in many aspects.

*As a biogeoscience paper, we think the manuscript falls disappointingly short of reaching some interesting potential for insight. Specifically, there is a substantial mismatch between data and the model for some very basic forest/ecosystem characteristics. There are large errors in LAI, GPP, RH, and age structure, as conceded by the authors. Even for the control site, GPP shows an almost opposite seasonality between model and data. The errors caused by calibration are much greater than the variation due to disturbance levels (Table 5). While it would have been preferable to have a more successful calibration, falling short of that, the authors should present a coherent and robust explanation of what the fundamental structural problems were, with figures specifically illustrating the insights. That would elevate the significance of the paper, and increase its utility for those seeking to do similar work.*

**Response:**

The low LAI bias is a characteristics of the verson of FATES in the original manuscript. In the revision, we have updated the model to a new version of FATES in which the penalty for establishing leaf biomass is greatly reduced. We also performed ensemble simulations to evaluate potential ways to show how key physiological parameters could influence the results. We have revised the manuscript to incorporate the new results and provide a summary of the ensemble simulations in the supplement.

***Comments for each section Introduction****: The authors lay out pertinent background information but the text does not explicitly articulate a cogent and compelling argument for why this study is needed. I think the introduction would be more effective if text were added to make the connection between the background information and the aims of the paper.*

**Response:**

Thanks for pointing this out. We have revised the introduction section to explicitly articulate the need to better represent wood harvest in next generation Earth system models, in which FATES will be a component in the revised manuscript.

***Methods:*** *They report that FATES is very sensitive to parameter values, to a point that with some combinations the two PFTs cannot coexist. That is somewhat worrisome. There is a fair amount of detail given on how a logging activity is applied to a patch, but it's a bit unclear which patches are selected for logging.*

**Response:**

In the revised manuscript, we acknowledge that ensuring co-existence continues to be an issue in FATES and we will try to improve it in newer versions. Nevertheless, parameterizations in the logging module do not require co-existence. Currently, we assumed that for a site such as km83, once logging is activated, trees will be harvested from all patches. We have added this information to the revised manuscript. We also added information on new developments in FATES where the time since disturbances is added prognostic variables to track the history of land use, key for applications of the model at regional to global scales.

***Results:*** *Given that SH mismatch happens at the seasonal scale it would be useful to have some analysis results at that time scale. For example, the results in Fig 4 could be replotted at the seasonal scale (average across years). Low SH is attributed to attributed to low LAI, but what's causing LAI? It seems a fairly straight forward question to answer (or at least speculate). It appears they did not go far enough with the most interesting/instructive part of the exploration. Similarly with soil moisture, the authors present a cursory analysis of soil water uptake. What about SWC of the deeper layer(s)? What is their relation to simulated ET?*

**Response**:

We have fixed the LAI problem in the revision, please check the revised manuscript to see if the updated results are satisfactory or not.

***Line-by-line comments***
*56: "suggested a net tropical forest land-use source of 1.3 : : :" is grammatically incorrect. I suggest something like, "suggested tropical forests can be a net source of 1.3… from land-use change."*
**Response**: done.

*63: The authors defined degradation as widespread damage to remaining trees, subcanopy vegetation and soils, and that it could cause as much as 40% carbon loss of clearcut deforestation. In your simulation, how did you define the effect of degradation?*

**Response**: The selective logging numerical experiments are meant to represent different levels of degradation. We will make this point clear in the revised manuscript.

*66: delete "as".*
**Response**: done.

*67: hyphenate one-eighth.*
**Response**: done.

*70: Extraneous parenthesis.*
**Response**: We have removed it.

*78: couple terrestrial and atmospheric: : :*
**Response**: done.

*78-80: Perhaps list some examples of those models?*
**Response**: Will do

*79: comma after "change"*
**Response**: done.

*81: representation of wood harvest: : :*
**Response**: done.

*83: Is that in LM3V? It's unclear.*
**Response**: Yes, we will clarify it.

*86-89: It would be better to define selective logging earlier, since it is referred to many times prior to this point in the text.*
**Response**: Yes, will have moved the definition to the first paragraph.

*90: Not just simplified but absent*
**Response**: will add that point.

*91: did not*
**Response**: done.

*98: "tremendous" is overly dramatic*
**Response**: modified to be "a lot"

*108: "assess the simulated recovery of Tapajos National Forest: : :"*
**Response**: done.

*109: summary of*
**Response**: done.

*113: simulated forest trajectory*
**Response**: done.

*120: The authors describe FATES model as a further developed model based on CLM(ED), which can be viewed as an early version of FATES. Then, which version is what you used in this study? Is there any paper that formerly published FATES model?*
**Response**: The model version used has been and will be provided in the Code and data availability section. A number of manuscripts are currently in various stages of review. We will list published/accepted ones in the revised manuscript.

*152: Specific should be lowercase. But lines 150-155 is a run-on sentence. They need to at least insert a conjunction.*
**Response**: done.

*162: Delete hyphen in co-existence*
**Response**: done.

*164: Delete hyphen in co-existence*
**Response**: done.

*176-7: "transports off-site by adding: : :" should be "transports off-site by reducing site carbon pools. Remaining necromass : : : are added to coarse woody debris and litter pools."*
**Response**: done.

*181: "are represented" should be "are conceptually represented." Because this paragraph just talks about the various concepts, and not specific implementation of logging regimes & effects.*
**Response**: done.

*189: It's unclear if FATES implements these two types of logging practices.*
**Response**: Yes, FATES is now able to represent these two practices by changing parameters in the logging module. We have clarified this in the revised manuscript.

*221-4: Parentheses seem unnecessary.*

**Response**: we have removed them.

*227: ": : :whose: : :" is grammatically incorrect.*
**Response**: changed to the. Thanks.

*250: Delete respectively.*
**Response**: done.

*273: Awkward phrasing. I recommend, "To … conservation, we calculate,". And then say how del-B is used to ensure mass conservation. (Just calculating del-B doesn't ensure mass conservation).*
**Response**: done.

*315: Equation should be plural. Or, if singular, use an article.*
**Response**: changed to plural.

[revised manuscript text omitted]

Following the existing CWD structure in FATES (*Fisher et al.*, 2015), CWD in the logging module is first separated into two categories: above-ground CWD and below-ground CWD. Within each category, four size classes are tracked based on their source, following Thonicke et al. (2010): trunks, large branches, small branches and twigs. Above-ground CWD from trunks ($\text{CWD}_{\text{trunk\_agb}}$, [kg C]) and large branches/small branches/twig ($\text{CWD}_{\text{branch\_agb}}$, [kg C]) are calculated as follows:

$$CWD_{\text{trunk\_agb}} = D_{\text{indirect}} \times B_{\text{stem\_agb}} \times f_{\text{trunk}} \times A \qquad (4)$$

$$CWD_{\text{branch\_agb}} = D_{\text{total}} \times B_{\text{stem\_agb}} \times f_{\text{branch}} \times A \qquad (5)$$

where $B_{\text{stem\_agb}}$ is the amount of above ground stem biomass in the cohort, $f_{\text{trunk}}$ and $f_{\text{branch}}$ represent the fraction of trunks and large branches/small branches/twig. Similarly, the below-ground CWD from trunks ($\text{CWD}_{\text{trunk\_bg}}$, [kg C]) and branches/twig ($\text{CWD}_{\text{branch\_bg}}$, [kg C]) are calculated as follows:

$$CWD_{\text{trunk\_bg}} = D_{\text{total}} \times B_{\text{root\_bg}} \times f_{\text{trunk}} \times A \qquad (6)$$

$$CWD_{\text{branch\_bg}} = D_{\text{total}} \times B_{\text{root\_bg}} \times f_{\text{branch}} \times A \qquad (7)$$

[revised manuscript text omitted]

---

## Author Response (AR2)

**Assessing impacts of selective logging on water, energy, and carbon budgets and ecosystem dynamics in Amazon forests using the Functionally Assembled Terrestrial Ecosystem Simulator [MS No.: bg-2019-129]**

**Responses to review comments**

*Associate Editor's comment:*
*I have now received a new referee report on your revised manuscript. This report is fairly positive but also raises a few questions. Please read this report carefully and revise your manuscript as needed in order to address the questions and comments from this referee.*

*Response:*
We greatly appreciate the associate editor and reviewers for their insightful and encouraging comments. We have made the suggested changes to the manuscript in response to these comments, and believe that the revised manuscript have addressed all of the reviewers' concerns.

Reviewer comments are shown in *Italic*. Author responses are shown in plain text.

*Anonymous Referee #2*

*The authors clearly improved the calibration since the previous version.*
**Response**: Thanks for your encouragement.

*In the response letter, the authors mentioned that they updated the model to a new version of FATES. This is not clear in section 2. Did you only reduce the penalty for establishing leaf biomass from the old version?*
**Response**: We did not emphasize the change between versions because it is a little strange to refer to an unaccepted version of the same manuscript and elaborate the version changes there. Nevertheless, the new simulations in the revised manuscript were performed using the FATES version https://github.com/NGEET/fates/releases/tag/sci.1.27.2_api.7.3.0, in which the penalty for establishing leaf biomass is significant reduced based. In the revised manuscript, the model version has been provided in the *Code and data availability* section with a user's guide to show how to check out the codes and run the simulations. The model version used in the original submission is archived at https://github.com/huangmy/fates-old/tree/logging-experiments, which is obsolete and no longer available in the FATES github repository.

Since the responses will be published with the paper, we hope that this response can serve as a clarification for readers who have confusions.

*In Table 1, the changes are an 0.001 increase in SLA for both PFTs, and significant increase in Vcmax for late successional PFT. Are these two increases the major reason for the better fitting in LAI and GPP? Or what is the major modification that leads to the improvement in model-data fitting?*
**Response**: In the original manuscript, the values of SLA are 0.016 and 0.015, those of Vcmax are 68 and 60, for the early and late successional PFTs respectively. In the revised manuscript, the values of SLA are reduced to 0.015 and 0.014, those of Vcmax are reduced to 65 and 50. Other parameter values remain unchanged. With such reductions in SLA and Vcmax, simulated GPP maintain at a similar level as that in the original manuscript, while LAI is significantly improved. Therefore, we believe that the model structural improvement in reducing for establishing leaf biomass was successful and is the key reason for the better model-data fitting.

*Moreover, compared with the previous version, why do NPP and RH lose their seasonality?*
**Response**: In the original manuscript, the comparison of NPP, HR, and AR were performed by comparing simulated monthly values against the annual estimates, which was not appropriate. In the revised version, we fixed this issue by comparing simulated annual values against estimated annual values from Miller et al., 2011. Such a difference is reflected in units of vertical axis in indicated in the units of the vertical axis in Figure 6 in the revised manuscript. We added a sentence in the caption of figure 6 of clarify it.

*Table S1 and Figure S1 were used to show the uncertainty range due to different parameter combinations. However, it is not clear how did you determined the final set of parameter values in Table 1. Did you choose the set of parameter values, which gave you the lowest RMSE of NEE or something else?*
**Response**: Thank you for asking this question. We selected the final set of parameter values using the following criteria:
1. Calculate the RMSEs of NEE, GPP, ER, AGB, CWD, and LAI of each simulation;
2. Select 10 simulations with the lowest sums of RMSEs of the selected variables;
3. Finalize the selection by choosing the simulations with the best ratio of early vs late successional PFTs as indicated in Figure 5.

We have added this procedure to the caption of Table S1 in the revised manuscript.

[revised manuscript text omitted]